# Removing Spurious Concepts from Neural Network Representations via Joint Subspace Estimation

## Abstract

Out-of-distribution generalization as well as the interpretation of neural networks is often hampered by spurious correlations. A common mitigation strategy is to remove spurious concepts from the neural network. Existing concept-removal methods tend to be overzealous by inadvertently eliminating features associated with the main task of the model. We propose an iterative algorithm that separates spurious from main-task concepts by jointly identifying two low-dimensional orthogonal subspaces in the neural network representation. Evaluated on benchmark datasets for computer vision and natural language processing, we show that it outperforms existing concept-removal methods, and is competitive with state-of-the-art out-of-distribution generalization techniques.

## 1 Introduction

Deep neural networks (DNNs) are typically trained to optimize their performance on a given dataset, thereby often relying on spurious correlations within the data (Gururangan et al., 2018; Srivastava et al., 2020; Wang & Culotta, 2020; Sagawa et al., 2020; Zhou et al., 2021). For example, if a model's main task is to distinguish between images of cows and penguins, the training data might contain a spurious correlation between animal type and background (cows typically appear on grasslands, penguins on snow). The model could exploit this correlation by basing its main-task classification on the representation of the background instead of the animal type (Geirhos et al., 2020). This is problematic in a variety of ways. For example, it can lead to a deterioration of model performance on out-of-distribution (OOD) data (Puli et al., 2022) and fool explainability methods (Kumar et al., 2022).

For tabular data, one can simply remove a variable if deemed spurious. However, removing spurious correlations directly from images or text is non-trivial and costly. An alternative is to focus on the embeddings, which are vector representations generated by the neural network of images or text. Post-hoc concept-removal methods aim to eliminate a concept from the embeddings, after the neural network has been trained. A *concept* refers to a representation in the data of a human-defined object or phenomenon (Kim et al., 2018). In the example of distinguishing cows and penguins, the background type is the *spurious concept*. Typically the parameters of the neural network are frozen and a concept classifier is trained on the embeddings, from which the concept features are then removed (Ravfogel et al., 2020; 2022a). Afterwards, a linear classifier is trained on the transformed embeddings to predict the *main-task concept* (e.g. animal type) and thereby preventing the model from using the spurious concept for main-task classification.

An application of concept-removal methods is OOD generalization in the presence of spurious correlations. They have important additional advantages relative to other OOD generalization techniques such as data augmentation (Hermann et al., 2020) and instance reweighing (Sagawa et al., 2020). For example, they can increase the interpretability of the model (Elazar et al., 2021), remove sensitive information (e.g. gender or race) from the neural network representations (Wang et al., 2019), or be used for transfer learning (Ravfogel et al., 2022a).

A drawback of post-hoc removal methods is that they tend to remove also main-task features from the embeddings, because due to the spurious correlation main-task features can be used to predict the spurious concept. For example, a concept classifier might use the cow's horns to predict a

grassland background. This has undesirable consequences, e.g. it hurts the main-task performance of the model (Ravfogel et al., 2020; Belinkov, 2022; Kumar et al., 2022), and limits the application of post-hoc removal methods in the most relevant cases, namely when the spurious correlation is strong and likely to be exploited by the model.

**Our contribution**: this paper improves on current post-hoc concept-removal methods by separating spurious and main-task concepts in the embedding space. We do so by jointly identifying two low-dimensional orthogonal subspaces, one associated with the spurious concept (e.g. background) and the other with the main-task concept (e.g. animal type). This crucially differs from existing methods, which only focus on the spurious concept features, risking the loss of vital main-task information. Furthermore, we make the identification of the subspaces systematic by introducing statistical tests that attribute directions in the embedding space to either the main-task or the spurious concept. The method, which we call Joint Subspace Estimation (JSE), is shown to be robust against the strength of the spurious correlation and to outperform existing concept-removal methods for a Toy dataset, as well as benchmark datasets for image recognition (Waterbirds, CelebA) and natural language processing (MultiNLI). Furthermore, we show that JSE is competitive with instance reweighing algorithms, which are state-of-the-art OOD generalization techniques not based on concept removal. We also validate that JSE is applicable when spurious concept labels are available for only a small subset of the data. A high-level overview of the method is given in Figure 1A.

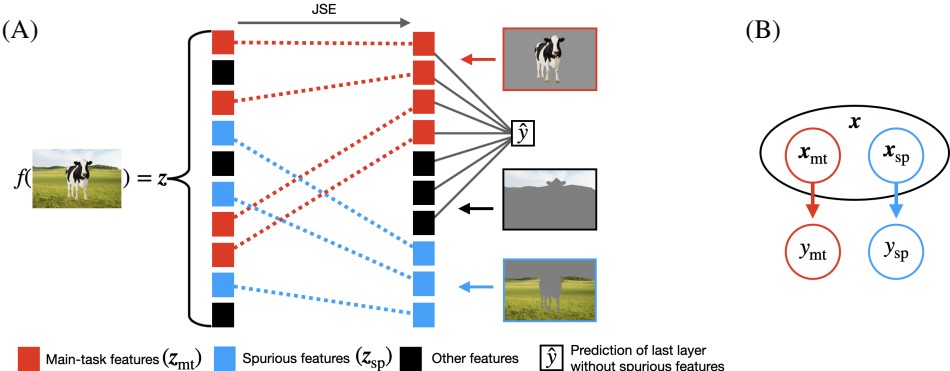

Figure 1: *(Panel A)* **Overview of Joint Subspace Estimation (JSE) for concept removal**: the input $x$ is fed through a neural network $f(x)$, from which we can extract the vector representation $z$. Within the vector representation, two orthogonal subspaces are identified: one related to the *spurious concept* (the background), and one to the *main-task concept* (bird type). JSE estimates the subspaces of the two concepts simultaneously to prevent mixing of spurious and main-task features. *(Panel B)* **Assumed causal relation between features and labels**: note that $x_{mt}$ and $x_{sp}$ can overlap, because specific pixels can contain information about both the main-task and spurious concept.

## 2 Spurious correlations and concept removal

We consider the random variables $\mathcal{D} = (y_{mt}, y_{sp}, x)$, where $y_{mt} \in \{0, 1\}$ is the main-task concept label, $y_{sp} \in \{0, 1\}$ is the spurious concept label and $x \in \mathcal{X}$ represents the input features. Each input $x$ contains subsets $x_{mt}$ and $x_{sp}$ of features corresponding to the main-task and spurious concept, respectively. In the example of cows and penguins, $x_{mt}$ and $x_{sp}$ correspond to the pixels showing the animal and the background. It is assumed that $x_{mt}$ and $x_{sp}$ causally determine the associated labels $y_{mt}$ and $y_{sp}$, respectively (see Figure 1B):

$$p\left(y_{mt}, y_{sp}, x_{mt}, x_{sp}\right) = p\left(y_{mt} | x_{mt}\right) p\left(y_{sp} | x_{sp}\right) p(x_{mt}, x_{sp}). \tag{1}$$

This implies $p(y_{mt} | x) = p(y_{mt} | x_{mt})$, but it does not mean that the main-task label $y_{mt}$ and the spurious features $x_{sp}$ are independent; they can be dependent due to dependence between $x_{mt}$ and $x_{sp}$. Since they are not causally related, we say they are spuriously correlated. At the level of trained neural networks, this means that a main-task classifier tends to make use of the spurious features $x_{sp}$ within $x$. Previous work offers a number of possible reasons, ranging from stochastic

gradient descent (SGD) training dynamics (Pezeshki et al., 2021) and overparameterization (Sagawa et al., 2020; D'Amour et al., 2020) to inductive biases of DNNs (Rahaman et al., 2019).

We now restrict our analysis to DNNs for classification, which typically consist of a complicated function $f(\boldsymbol{x}) : \mathcal{X} \to \mathbb{R}^d$ mapping the input features to a vector representation, followed by a linear layer. We assume the embedding vectors $\boldsymbol{z} \in \mathbb{R}^d$ have a similar structure as the input features $\boldsymbol{x}$, in the sense that each $\boldsymbol{z}$ has subsets $\boldsymbol{z}_{\mathrm{mt}} \in \mathcal{Z}_{\mathrm{mt}} \subseteq \mathbb{R}^d$ and $\boldsymbol{z}_{\mathrm{sp}} \in \mathcal{Z}_{\mathrm{sp}} \subseteq \mathbb{R}^d$ that causally determine the labels $y_{\mathrm{mt}}$ and $y_{\mathrm{sp}}$, respectively. It should be stressed that this does not necessarily hold. The trained DNN could have mixed the different features, or have one of them removed, because of their mutual predictive ability for the main-task label $y_{\mathrm{mt}}$. However, there is recent empirical evidence that even when trained on data with a spurious correlation, neural networks tend to learn both main-task and spurious features (Kirichenko et al., 2023; Izmailov et al., 2022; Rosenfeld et al., 2022).

In addition, we assume that $\mathcal{Z}_{\mathrm{mt}}$ and $\mathcal{Z}_{\mathrm{sp}}$ are linear subspaces of the embedding space $\mathbb{R}^d$. This sometimes goes under the name of *linear subspace hypothesis* (Bolukbasi et al., 2016; Vargas & Cotterell, 2020). Previous work shows that linear subspaces can encode information about complex concepts (Bau et al., 2017; Alain & Bengio, 2017). Moreover, non-linear information about the spurious concept cannot be used by the last layer for binary classification (Ravfogel et al., 2023).

One concept-removal approach to the problem of OOD generalization is to project the embedding vectors $\boldsymbol{z} \in \mathbb{R}^d$ onto a linear subspace, before feeding it to the final linear layer. Suppose that $\boldsymbol{v}_{\mathrm{sp},1}, \boldsymbol{v}_{\mathrm{sp},2}, \dots, \boldsymbol{v}_{\mathrm{sp},d_{\mathrm{sp}}}$ is an orthonormal basis of the spurious embedding subspace $\mathcal{Z}_{\mathrm{sp}} \subseteq \mathbb{R}^d$ and $\boldsymbol{V}_{\mathrm{sp}}$ is the matrix $(d \times d_{\mathrm{sp}})$ whose columns are the basis vectors. Then the transformation from $\boldsymbol{z}$ to $(\boldsymbol{I} - \boldsymbol{V}_{\mathrm{sp}} \boldsymbol{V}_{\mathrm{sp}}^T) \boldsymbol{z}$ is the orthogonal projection onto the orthogonal complement of $\mathbb{R}^d \setminus \mathcal{Z}_{\mathrm{sp}}$ and thereby removes the spurious features from the representation. A linear layer that uses the transformed embeddings to predict the binary main-task label $y_{\mathrm{mt}}$ will therefore not use the spurious concept and model performance will not be affected when applied to OOD data.

In practice, however, it is highly non-trivial to estimate (a basis of) the subspace $\mathcal{Z}_{\mathrm{sp}}$. Due to the spurious correlations, classifiers that use the embedding vectors to predict the spurious label $y_{\mathrm{sp}}$ also make use of the main-task embeddings $\boldsymbol{z}_{\mathrm{mt}}$. As a consequence, an estimate of $\mathcal{Z}_{\mathrm{sp}}$ will also contain directions that are actually part of $\mathcal{Z}_{\mathrm{mt}}$. Projecting out the estimate of $\mathcal{Z}_{\mathrm{sp}}$ removes main-task information and therefore hurts the performance of the resulting main-task classifier (Ravfogel et al., 2020; Belinkov, 2022; Kumar et al., 2022). We will see that our JSE method addresses this problem by estimating not only $\mathcal{Z}_{\mathrm{sp}}$, but also the main-task embedding space $\mathcal{Z}_{\mathrm{mt}}$.

## 2.1 RELATED WORK

**Spurious correlations**: The problem of neural networks relying on spurious correlations has arisen in both computer vision (Geirhos et al., 2019; Xiao et al., 2021; Singla & Feizi, 2022) and NLP (Niven & Kao, 2019; Kaushik & Lipton, 2018; McCoy et al., 2019). There is a wide range of methods addressing spurious correlations in neural networks, including data augmentation (Hermann et al., 2020), invariant learning (Arjovsky et al., 2019; Ahuja et al., 2021), or instance reweighing (Sagawa et al., 2020). The latter category is most akin to concept-removal methods, as it uses (limited) availability of spurious concept labels. Three particularly powerful methods are: (i) group distributional robust optimization (GDRO), which aims to minimize the worst-group loss over possible combinations of the spurious and main-task labels (Sagawa et al., 2019), (ii) just train twice (JTT), which puts a greater weight on samples that were wrongly identified by an initial model (Liu et al., 2021), and (iii) group-weighted empirical risk minimization (GW-ERM), which uses a sampling scheme such that the spurious concept and main-task labels are balanced (Idrissi et al., 2022). This latter method is particularly effective when re-training the last layer (Kirichenko et al., 2023).

**Concept-removal methods**: Concept removal is mainly based on adversarial approaches (Goodfellow et al., 2014), commonly to mitigate undesirable biases (Edwards & Storkey, 2016; Zhang et al., 2018; Wang et al., 2021). This is frequently referred to as adversarial removal (ADV). However, the ability of ADV to remove concepts has been called into question (Elazar & Goldberg, 2018). An alternative is to remove a linear subspace from the embeddings (Bolukbasi et al., 2016; Ethayarajh et al., 2019; Dev & Phillips, 2019; Dev et al., 2021). A key method in this category is iterative null-space projection (INLP, Ravfogel et al., 2020), in which a linear classifier predicts the concept labels, and the coefficients of the classifier are orthogonally projected from the embeddings. This

is repeated until the concept cannot be predicted. A follow-up method is relaxed linear adversarial concept erasure (RLACE), in which an orthogonal projection matrix is trained such that the concept cannot be predicted from the embeddings (Ravfogel et al., 2022a).

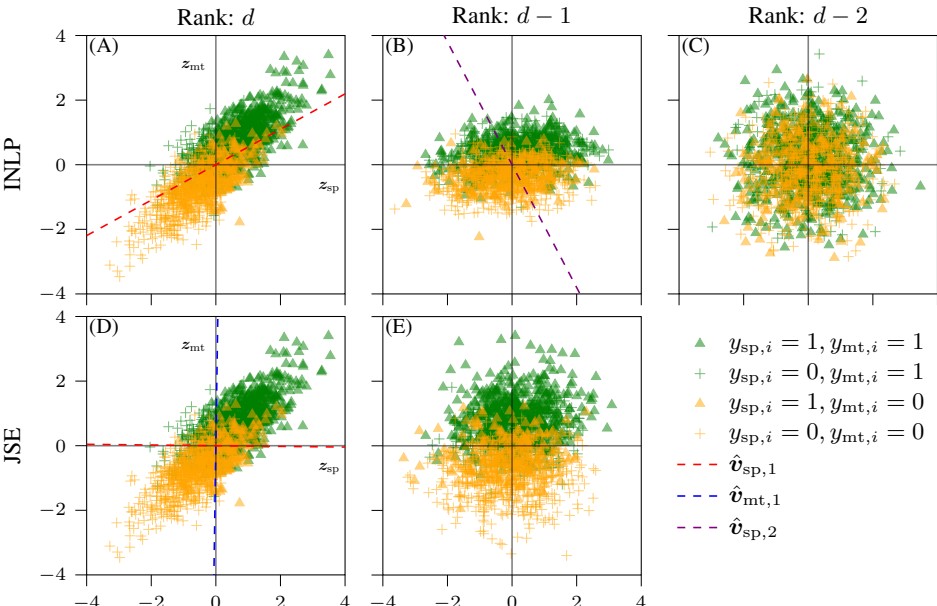

Figure 2: **Illustration of JSE, in comparison to INLP**: based on the $d(=20)$-dimensional Toy dataset (see Section 4.1) with $\rho = 0.8$ and sample size $n = 2,000$. Two-dimensional slices of the embedding space are shown. Panels A and D have the spurious feature on the x-axis and the main-task feature on the y-axis. The remaining panels show the axes that best separate the main-task labels. JSE identifies the main-task and spurious directions (panel D) and the remaining class separation is attributed to the main-task concept (panel E). INLP identifies (superpositions of) the main-task and spurious directions as spurious (panels A and B), and the main-task information is removed (panel C).

## 3 JOINT SUBSPACE ESTIMATION

We will now introduce Joint Subspace Estimation (JSE) in which the spurious and main-task embedding subspaces $\mathcal{Z}_{\text{sp}}$ and $\mathcal{Z}_{\text{mt}}$ are estimated simultaneously. Section 3.1 explains how to simultaneously estimate individual basis vectors for $\mathcal{Z}_{\text{sp}}$ and $\mathcal{Z}_{\text{mt}}$. Section 3.2 introduces an iterative procedure to find multiple basis vectors for $\mathcal{Z}_{\text{sp}}$ and $\mathcal{Z}_{\text{mt}}$. Section 3.3 puts forward two statistical tests to terminate the iterative procedure and to determine the dimensions of $\mathcal{Z}_{\text{sp}}$ and $\mathcal{Z}_{\text{mt}}$.

### 3.1 ESTIMATING SPURIOUS AND MAIN-TASK CONCEPT VECTORS

As a starting point, consider simultaneously estimating one vector $\boldsymbol{v}_{\text{sp}} \in \mathcal{Z}_{\text{sp}}$ and another vector $\boldsymbol{v}_{\text{mt}} \in \mathcal{Z}_{\text{mt}}$. A usual approach for estimating $\boldsymbol{v}_{\text{sp}}$ is to train a logistic regression on the embeddings, $\hat{y}_{\text{sp}} = \text{Logit}^{-1}(\boldsymbol{z}^T \boldsymbol{w}_{\text{sp}} + b_{\text{sp}})$, and then to use the (normalized) coefficients $\boldsymbol{v}_{\text{sp}} = \boldsymbol{w}_{\text{sp}}/\|\boldsymbol{w}_{\text{sp}}\|$ as a so-called concept vector $\boldsymbol{v}_{\text{sp}}$ that contains information about the concept (Kim et al., 2018). However, if we perform logistic regression in a sample where the spurious and main-task features are correlated, then the estimate of $\boldsymbol{v}_{\text{sp}}$ might have components in the direction of main-task features (and vice versa for the estimate of $\boldsymbol{v}_{\text{mt}}$). To address this issue, we need to assume a relation between the two subspaces, which is done by the following assumption.

**Orthogonality Assumption.** *The linear subspaces $\mathcal{Z}_{\text{sp}}$ and $\mathcal{Z}_{\text{mt}}$ are orthogonal, i.e. each vector $\boldsymbol{v}_{\text{sp}} \in \mathcal{Z}_{\text{sp}}$ is perpendicular to each vector $\boldsymbol{v}_{\text{mt}} \in \mathcal{Z}_{\text{mt}}$.*

This is consistent with the assumption of the features determining the labels $y_{\text{sp}}$ and $y_{\text{mt}}$ being distinct, and the empirical observation that high-level concepts are distinctly represented in the embed-

dings (Kirichenko et al., 2023). Also note that orthogonality does not imply independence between main-task and spurious features, as additionally assumed in earlier work (Chen et al., 2020).

The idea behind the orthogonality assumption is that it discourages the estimate of $\boldsymbol{v}_{\mathrm{sp}}$ to use main-task features, as it is forced to be perpendicular to a main-task direction $\boldsymbol{v}_{\mathrm{mt}}$, and vice-versa. We illustrate the effect of the assumption in Figure 2 (panels A and D) for Toy data, and analyse how JSE is affected if the orthogonality assumption does not hold in Appendix B. An alternative perspective on the orthogonality assumption is that we aim to identify subspaces of $\mathcal{Z}_{\mathrm{sp}}$ and $\mathcal{Z}_{\mathrm{mt}}$ that are orthogonal to each other and that are informative about the respective labels. If $\mathcal{Z}_{\mathrm{sp}}$ and $\mathcal{Z}_{\mathrm{mt}}$ are high-dimensional (applicable in most realistic settings), there are more degrees of freedom for these subspaces to cover significant parts of $\mathcal{Z}_{\mathrm{sp}}$ and $\mathcal{Z}_{\mathrm{mt}}$ in terms of their ability to predict $y_{\mathrm{sp}}$ and $y_{\mathrm{mt}}$.

We thus simultaneously perform a logistic regression on the embeddings $\boldsymbol{z}$ for $y_{\mathrm{sp}}$ and $y_{\mathrm{mt}}$, subject to the constraint of orthogonality of $\boldsymbol{w}_{\mathrm{sp}}$ and $\boldsymbol{w}_{\mathrm{mt}}$. This means that for a sample $\{y_{\mathrm{mt},i}, y_{\mathrm{sp},i}, \boldsymbol{z}_i\}_{i=1}^n$ we perform the following optimization,

$$\hat{\boldsymbol{w}}_{\mathrm{sp}}, \hat{\boldsymbol{w}}_{\mathrm{mt}}, \hat{b}_{\mathrm{sp}}, \hat{b}_{\mathrm{mt}} = \operatorname*{arg\,min}_{\substack{\boldsymbol{w}_{\mathrm{sp}}, \boldsymbol{w}_{\mathrm{mt}}, b_{\mathrm{sp}}, b_{\mathrm{mt}} \\ (\boldsymbol{w}_{\mathrm{sp}} \perp \boldsymbol{w}_{\mathrm{mt}})}} \sum_{i=1}^n \mathcal{L}_{\mathrm{BCE}}(\hat{y}_{\mathrm{sp},i}, y_{\mathrm{sp},i}) + \mathcal{L}_{\mathrm{BCE}}(\hat{y}_{\mathrm{mt},i}, y_{\mathrm{mt},i}), \qquad (2)$$

where $\mathcal{L}_{\mathrm{BCE}}$ is the binary cross-entropy (BCE). Furthermore, $\hat{y}_{\mathrm{sp},i} = \mathrm{Logit}^{-1}\left(\boldsymbol{z}_i^T \boldsymbol{w}_{\mathrm{sp}} + b_{\mathrm{sp}}\right)$, and similarly for $\hat{y}_{\mathrm{mt},i}$. The estimated spurious and main-task concept vectors are then $\hat{\boldsymbol{v}}_{\mathrm{sp}} = \hat{\boldsymbol{w}}_{\mathrm{sp}}/\|\hat{\boldsymbol{w}}_{\mathrm{sp}}\|$ and $\hat{\boldsymbol{v}}_{\mathrm{mt}} = \hat{\boldsymbol{w}}_{\mathrm{mt}}/\|\hat{\boldsymbol{w}}_{\mathrm{mt}}\|$.

## 3.2 Iteratively estimating multiple concept and main-task vectors

The concept subspaces $\mathcal{Z}_{\mathrm{sp}}$ and $\mathcal{Z}_{\mathrm{mt}}$ will generally not be one-dimensional. As a consequence, the estimated spurious concept vector $\hat{\boldsymbol{v}}_{\mathrm{sp}}$ could still contain main-task components (and vice versa for $\hat{\boldsymbol{v}}_{\mathrm{mt}}$). To address this, we propose an iterative procedure to estimate orthonormal bases of the subspaces $\mathcal{Z}_{\mathrm{sp}}$ and $\mathcal{Z}_{\mathrm{mt}}$, which are guaranteed to be orthogonal to each other.

---

**Algorithm 1** JSE algorithm to estimate orthonormal bases for $\mathcal{Z}_{\mathrm{sp}}$ and $\mathcal{Z}_{\mathrm{mt}}$. The conditions in the **if**-statements are discussed in Section 3.3.

---

**Require:** a sample $\{y_{\mathrm{mt},i}, y_{\mathrm{sp},i}, \boldsymbol{z}_i\}_{i=1}^n$ consisting of two binary labels and a vector $\boldsymbol{z}_i \in \mathbb{R}^d$.

Initialize a $(n \times d)$-dimensional embedding matrix $\boldsymbol{Z} = (\boldsymbol{z}_1\, \boldsymbol{z}_2\, \cdots\, \boldsymbol{z}_n)^T$.
Initialize $\boldsymbol{Z}_{\mathrm{sp}}^{\perp} \leftarrow \boldsymbol{Z}$.
**for** $i = 1, ..., d$ **do**
  $\boldsymbol{Z}_{\mathrm{remain}} \leftarrow \boldsymbol{Z}_{\mathrm{sp}}^{\perp}$
  **for** $j = 1, ..., d$ **do**
    Estimate $\hat{\boldsymbol{w}}_{\mathrm{sp}}$ and $\hat{\boldsymbol{w}}_{\mathrm{mt}}$ with Equation 2, using embeddings $\boldsymbol{Z}_{\mathrm{remain}}$.
    Define normalized directions $\hat{\boldsymbol{v}}_{\mathrm{sp},i} \leftarrow \hat{\boldsymbol{w}}_{\mathrm{sp}}/\|\hat{\boldsymbol{w}}_{\mathrm{sp}}\|$ and $\hat{\boldsymbol{v}}_{\mathrm{mt},j} \leftarrow \hat{\boldsymbol{w}}_{\mathrm{mt}}/\|\hat{\boldsymbol{w}}_{\mathrm{mt}}\|$.
    **if** $\hat{\boldsymbol{v}}_{\mathrm{mt},j}$ is a proper main-task direction **then**
      Projection $\boldsymbol{Z}_{\mathrm{remain}} \leftarrow \boldsymbol{Z}_{\mathrm{sp}}^{\perp}(\boldsymbol{I} - \hat{\boldsymbol{V}}_{\mathrm{mt}}\hat{\boldsymbol{V}}_{\mathrm{mt}}^T)$, where $\hat{\boldsymbol{V}}_{\mathrm{mt}} = (\hat{\boldsymbol{v}}_{\mathrm{mt},1}\, \hat{\boldsymbol{v}}_{\mathrm{mt},2}\, \cdots\, \hat{\boldsymbol{v}}_{\mathrm{mt},j})$.
    **else**
      **break**
    **end if**
  **end for**
  **if** $\hat{\boldsymbol{v}}_{\mathrm{sp},i}$ is a proper spurious direction **then**
    Projection $\boldsymbol{Z}_{\mathrm{sp}}^{\perp} \leftarrow \boldsymbol{Z}(\boldsymbol{I} - \hat{\boldsymbol{V}}_{\mathrm{sp}}\hat{\boldsymbol{V}}_{\mathrm{sp}}^T)$, where $\hat{\boldsymbol{V}}_{\mathrm{sp}} = (\hat{\boldsymbol{v}}_{\mathrm{sp},1}\, \hat{\boldsymbol{v}}_{\mathrm{sp},2}\, \cdots\, \hat{\boldsymbol{v}}_{\mathrm{sp},i})$.
  **else**
    **break**
  **end if**
**end for**
**return** $\hat{\boldsymbol{v}}_{\mathrm{sp},1},\, \hat{\boldsymbol{v}}_{\mathrm{sp},2},\, \ldots,\, \hat{\boldsymbol{v}}_{\mathrm{sp},i}$

---

For now, let us focus on estimating a vector $\boldsymbol{v}_{\mathrm{sp}} \in \mathcal{Z}_{\mathrm{sp}}$. By applying the procedure of Equation 2 gives a $\hat{\boldsymbol{v}}_{\mathrm{sp}}$ and $\hat{\boldsymbol{v}}_{\mathrm{mt}}$, where $\hat{\boldsymbol{v}}_{\mathrm{sp}}$ may still have components in $\mathcal{Z}_{\mathrm{mt}}$ and $\hat{\boldsymbol{v}}_{\mathrm{mt}}$ may have components in $\mathcal{Z}_{\mathrm{sp}}$. We propose to project out the direction $\hat{\boldsymbol{v}}_{\mathrm{mt}}$ from the embeddings and to repeat the optimization

of Equation 2 for the resulting subspace. Doing this multiple times will eventually remove all main-task information from the subspace, guaranteeing that the estimated vector $\hat{\boldsymbol{v}}_{\mathrm{sp},1}$ is orthogonal to the (estimated) main-task subspace.

By projecting out $\hat{\boldsymbol{v}}_{\mathrm{sp},1}$ from the original embeddings and repeating the whole procedure $d_{\mathrm{sp}} = \dim(\mathcal{Z}_{\mathrm{sp}})$ times, we estimate an orthonormal basis $\hat{\boldsymbol{v}}_{\mathrm{sp},1}, \hat{\boldsymbol{v}}_{\mathrm{sp},2}, \ldots, \hat{\boldsymbol{v}}_{\mathrm{sp},d_{\mathrm{sp}}}$ of $\mathcal{Z}_{\mathrm{sp}}$ that is orthogonal to the (main-task) subspace. The method described is a nested for-loop (see Algorithm 1), where the inner loop finds and projects out main-task vectors, and the outer loop finds and projects out spurious vectors. After having found the basis of $\mathcal{Z}_{\mathrm{sp}}$, one can repeat the inner loop one last time to find $d_{\mathrm{mt}} = \dim(\mathcal{Z}_{\mathrm{mt}})$ vectors $\hat{\boldsymbol{v}}_{\mathrm{mt},1}, \hat{\boldsymbol{v}}_{\mathrm{mt},2}, \ldots, \hat{\boldsymbol{v}}_{\mathrm{mt},d_{\mathrm{mt}}}$ constituting an estimated basis for $\mathcal{Z}_{\mathrm{mt}}$. For a more detailed description of the algorithm, see Appendix C.

So far, we have treated the subspaces $\mathcal{Z}_{\mathrm{sp}}$ and $\mathcal{Z}_{\mathrm{mt}}$ equally. This symmetry is broken in Algorithm 1, as the main-task directions are identified in the inner loop and the concept directions in the outer loop. In Appendix C we give empirical evidence that swapping the inner and outer loop has little effect on the outcome of the JSE method. Furthermore, we show that training the linear classifier on $\mathcal{Z}_{\mathrm{mt}}$ instead of $\mathcal{Z}_{\mathrm{sp}}^{\perp}$ gives similar performance. Also, note that the computational cost of the double for-loop is limited, as the number of required light-weight optimizations (Equation 2) is at most $d_{\mathrm{mt}} \times d_{\mathrm{sp}}$. In the experiments performed in Section 4, the dimensions of the respective subspaces were found to be never larger than 10.

### 3.3 TESTING WHEN TO STOP ADDING CONCEPT OR MAIN-TASK VECTORS

So far, in the description of the iterative algorithm we have assumed the dimensions $d_{\mathrm{sp}}$ and $d_{\mathrm{mt}}$ of the respective subspaces $\mathcal{Z}_{\mathrm{sp}}$ and $\mathcal{Z}_{\mathrm{mt}}$ to be known. In practice, the dimensions must be estimated via stopping criteria of the (nested) for loops in Algorithm 1. Let us focus on the condition in the outer loop. The condition in the inner loop is, *mutatis mutandis*, the same. For a given (normalized) direction $\boldsymbol{v}_{\mathrm{sp}} \in \mathbb{R}^d$ in the embedding space, the statement "$\boldsymbol{v}_{\mathrm{sp}}$ is a proper spurious direction" means that two criteria are met:

1. **The direction $\boldsymbol{v}_{\mathrm{sp}}$ is informative about the spurious label** $y_{\mathrm{sp}}$, meaning that the embeddings projected onto $\boldsymbol{v}_{\mathrm{sp}}$ are able to predict the spurious label. To be concrete, a logistic regression based on the projected embeddings should have a higher accuracy than an classifier that just predicts the majority class, which we refer to as a 'random classifier'.

2. **The direction $\boldsymbol{v}_{\mathrm{sp}}$ should be more predictive of the spurious concept than of the main-task concept.** Due to the spurious correlation, a vector in $\mathcal{Z}_{\mathrm{sp}}$ is likely also predictive for the main-task concept. We nonetheless associate it with the spurious subspace $\mathcal{Z}_{\mathrm{sp}}$, as long as its prediction accuracy for the spurious label is higher than for the main-task label.

Note that the first criterion is already used in Ravfogel et al. (2020), while the second is novel and addresses the problem of inadvertently removing main-task information in existing concept-removal methods. This is illustrated in Figure 2 (panel B), where the INLP-method of Ravfogel et al. (2020) removes a feature that is more predictive of the main-task concept than the spurious concept.

To make these criteria operational, we introduce two statistical tests in terms of differences between BCE's. For the first criterion we compare the BCE of $\hat{y}_{\mathrm{sp}}^{(\boldsymbol{v}_{\mathrm{sp}})} = \mathrm{Logit}^{-1}\left(\gamma_{\mathrm{sp}}\boldsymbol{z}^T\boldsymbol{v}_{\mathrm{sp}} + b_{\mathrm{sp}}\right)$, which is a predictor for the label $y_{\mathrm{sp}}$ based on the embeddings projected onto $\boldsymbol{v}_{\mathrm{sp}}$, and the BCE of a majority-rule 'random classifier'. The model parameters $\gamma_{\mathrm{sp}}$ and $b_{\mathrm{sp}}$ are to be trained by minimizing the BCE. For the second criterion we compare the BCE of $\hat{y}_{\mathrm{sp}}^{(\boldsymbol{v}_{\mathrm{sp}})}$ with the analogously defined $\hat{y}_{\mathrm{mt}}^{(\boldsymbol{v}_{\mathrm{sp}})}$, which is a predictor of $y_{\mathrm{mt}}$. Both tests are performed using a $t$-statistic, using an equally weighted average of the BCE's over the four combinations of $y_{\mathrm{sp}}$ and $y_{\mathrm{mt}}$. For a precise definition of the hypotheses, test statistics, and their properties, see Appendix D.

## 4 EXPERIMENTS

We present two sets of experiments. First, we compare JSE with other last-layer concept-removal methods mentioned in Section 2.1: iterative null-space projection (INLP), relaxed linear adversarial concept erasure (RLACE) and adversarial removal based on a single linear adversary (ADV). To assess their ability to identify spurious features, we consider the problem of OOD generalization

to (Toy, vision and text) data with a different dependence between main-task and spurious features (i.e. $p_{\text{train}}(\boldsymbol{x}_{\text{mt}}, \boldsymbol{x}_{\text{sp}}) \neq p_{\text{OOD}}(\boldsymbol{x}_{\text{mt}}, \boldsymbol{x}_{\text{sp}})$, while the conditionals in Equation 1 and all marginal distributions remain the same). This generally leads to $p_{\text{train}}(y_{\text{mt}}|\boldsymbol{x}_{\text{sp}}) \neq p_{\text{OOD}}(y_{\text{mt}}|\boldsymbol{x}_{\text{sp}})$, with similar discrepancies at the level of the embeddings. A model using the spurious features to predict the main-task label will see a strong deterioration in performance when applied to the OOD data.

We then compare JSE with instance reweighing techniques mentioned in Section 2.1: just train twice (JTT), group distributional robust optimization (GDRO) and group-weighted ERM (GW-ERM). For a fair comparison in terms of computational costs, all methods are restricted to last-layer re-training.

The experiments presented in this section assume availability of spurious concept labels $y_{\text{sp}}$ for all datapoints. See Appendix E for empirical evidence that JSE also performs well with a limited set of spurious concept labels. Details about the datasets, experimental setup and parameter selection can be found in Appendix F. For numerical details, see Appendix A.

## 4.1 DATASETS

**Toy data:** We create $d-$dimensional embeddings drawn from a multivariate normal distribution with a block correlation matrix,

$$\boldsymbol{z} \sim \mathcal{N}(\boldsymbol{\mu} = \boldsymbol{0}, \boldsymbol{\Sigma}), \quad \text{where} \quad \boldsymbol{\Sigma} = \begin{bmatrix} \boldsymbol{\Sigma}_{\text{sp,mt}} & \boldsymbol{0} \\ \boldsymbol{0} & \boldsymbol{I} \end{bmatrix}, \quad \boldsymbol{\Sigma}_{\text{sp,mt}} = \begin{bmatrix} 1 & \rho \\ \rho & 1 \end{bmatrix}.$$

Note that, unlike more realistic situations, the embeddings here are not neural network representations of underlying input features. We set $\mathcal{Z}_{\text{sp}}$ and $\mathcal{Z}_{\text{mt}}$ to be one-dimensional, with spurious and main-task directions given by $\boldsymbol{w}_{\text{sp}} = (\gamma_{\text{sp}}, 0, 0, \ldots, 0)^T$ and $\boldsymbol{w}_{\text{mt}} = (0, \gamma_{\text{mt}}, 0, \ldots, 0)^T$, respectively. We define binary labels $y_{\text{sp}}$ and $y_{\text{mt}}$ following a logit model,

$$p(y_{\text{sp}} = 1|\boldsymbol{z}) = \text{Logit}^{-1}\left(\boldsymbol{z}^T \boldsymbol{w}_{\text{sp}} + b_{\text{sp}}\right), \quad p(y_{\text{mt}} = 1|\boldsymbol{z}) = \text{Logit}^{-1}\left(\boldsymbol{z}^T \boldsymbol{w}_{\text{mt}} + b_{\text{mt}}\right).$$

Throughout the simulations we take $d = 20$, $b_{\text{sp}} = b_{\text{mt}} = 0$ and $\gamma_{\text{sp}} = \gamma_{\text{mt}} = 3$. In this setup the parameter $\rho$ is the correlation between the spurious and main-task features, thereby determining the spurious relation $p_{\text{train}}(y_{\text{mt}}|\boldsymbol{z}_{\text{sp}})$ between the main-task label and the spurious feature.

**Vision:** We use two common datasets containing a spurious correlation. The first is the Waterbirds dataset (Sagawa et al., 2019), where the main-task concept is bird type (waterbird vs. landbird) and the spurious concept is background (water vs. land). The second is the CelebA dataset, where the main-task concept is hair color (blond vs. non-blond) and the spurious concept is sex (female vs. male). We use a pre-trained Resnet50 architecture (He et al., 2016) without finetuning, which is not needed for strong performance on either dataset (Izmailov et al., 2022).

**NLP:** We use the MultiNLI dataset (Williams et al., 2018), which contains pairs of sentences. The main-task concept is whether or not the first sentence contradicts the second sentence. Following an experiment from Joshi et al. (2022), we use as spurious concept the presence or absence of a punctuation mark ('!!') at the end of the second sentence. For exemplary pairs of sentences, see Appendix F. Each run in our experiments starts with finetuning a BERT model, after which concept removal is applied to the [CLS] embeddings.

For the vision and NLP datasets, we cannot directly control the strength of the spurious relation between the main-task label and the spurious features, $p_{\text{train}}(y_{\text{mt}}|\boldsymbol{z}_{\text{sp}})$. We therefore use $p_{\text{train}}(y_{\text{mt}} = y|y_{\text{sp}} = y)$, with $y \in \{0, 1\}$, as a proxy. To increase the precision of our method and for computational efficiency, we reduce the dimension of the last-layer embeddings to $d = 100$ via Principal Component Analysis (PCA).

## 4.2 COMPARISON WITH CONCEPT-REMOVAL METHODS

Figure 3 shows the results of applying different concept-removal methods to the problem of OOD generalization. For all datasets JSE outperforms the other methods, in particular when the spurious correlation is strong. The performance of INLP and RLACE decreases as they remove main-task features together with the spurious features. This behaviour is illustrated for the Toy dataset in Figure 2. For ADV the performance deteriorates because the spurious features remain present after

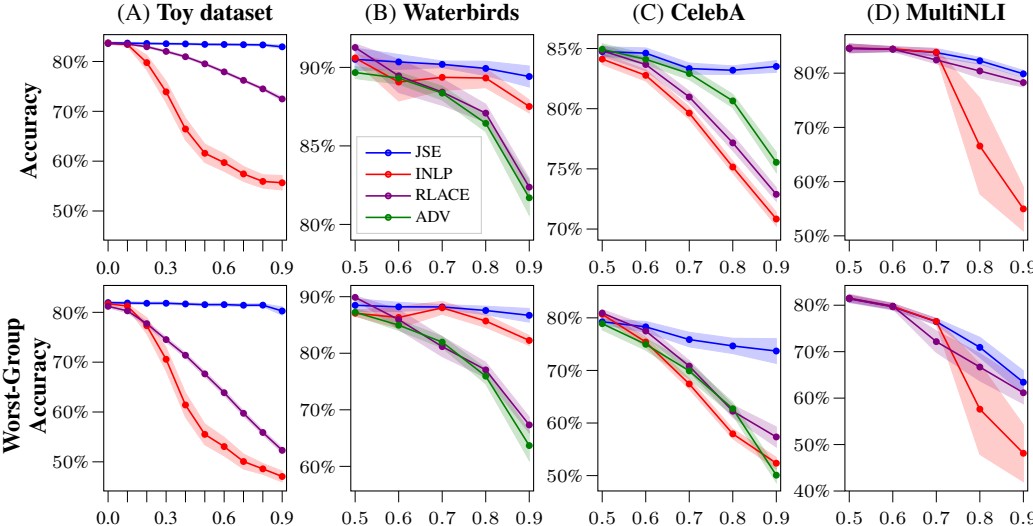

Figure 3: **OOD generalization, compared to other concept-removal methods**: We plot the (worst-group) accuracy on a test set without spurious correlation, as a function of the spurious correlation in the training set ($\rho$ for the Toy dataset, $p_{\text{train}}(y_{\text{mt}} = y | y_{\text{sp}} = y)$ for the other datasets). Averages based on 100, 20, 20 and 5 runs, respectively. The shaded area reflects the 95% confidence interval.

the training procedure, in line with previous work (Belinkov, 2022; Ravfogel et al., 2022a). Although much smaller, JSE also shows performance loss. As the spurious correlation increases, we expect our method to become more sensitive to finite-sample estimation noise. We illustrate this further in Appendix B.2.

The performance of JSE is particularly strong for the vision datasets. For the NLP dataset JSE exhibits a more significant performance loss for strong spurious correlations. We suspect that during finetuning of BERT the main-task concept and spurious concept become overlapping in the [CLS] embedding, in line with observations from previous work by Dalvi et al. (2022). The performance of RLACE is close to JSE's, although it appears to be mixing spurious and main-task features, as it did for the vision datasets. Attempts to perform adversarial removal on the MultiNLI dataset leads to convergence issues, which has been previously noted as a problem for these types of methods (Xing et al., 2021).

From a different perspective, JSE attempts to make a neural network focus on the right (i.e. causally related) features. Figure 4 shows this using Grad-CAM (Selvaraju et al., 2017). Although all trained to predict bird type, only for JSE the classifier relies predominantly on the bird features and neglects the background. Interestingly, both INLP and RLACE perform much worse on images that appear more frequently in the training set (e.g. landbirds on land) than on images from minority groups (e.g. landbirds on water). We posit that this is because features get mixed, as described by Kumar et al. (2022), leading INLP or RLACE to

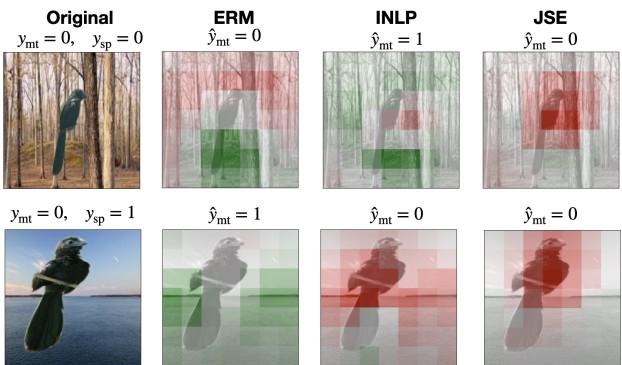

Figure 4: **Grad-CAM for the last layer of Resnet-50 predicting the main-task label**: Red (green) patches indicate a contribution towards the correct (incorrect) predicted class. ERM and INLP use the background for both their correct and incorrect predictions.

associate spurious features with the main-task concept. This can also be seen in Figure 4, where after INLP a landbird is classified as a waterbird with use of the land background.

Overall, the results indicate that JSE outperforms other concept-removal methods in identifying spurious features in the last layer, while preserving main-task features.

### 4.3 Comparison with instance reweighing techniques for OOD generalization

Figure 5 compares JSE to instance reweighing methods, the purpose of which specifically is OOD generalization in the presence of spurious correlations. We emphasize a conceptual difference between JSE and these methods; JSE removes the spurious embedding, which makes it robust to changes at the more fundamental level of the conditional distribution $p(y_{mt}|z_{sp})$. The other methods aim for robustness against changes in $p(y_{mt}|y_{sp})$. This explains why for the Toy dataset JSE outperforms the other methods, because in this setting it is the conditional distribution $p(y_{mt}|z_{sp})$ itself that is modified for the OOD data.

For the other datasets JSE outperforms JTT and is competitive with GW-ERM and GDRO. The sudden drop in (worst-group) accuracy for $p_{train}(y_{mt} = y|y_{sp} = y) = 0.95$ is due to finite-sample estimation noise (see Appendix B.2). We also note that, in contrast with the instance reweighing methods, the hyperparameters of JSE have been optimized for identifying the main-task and spurious concepts, and not for OOD generalization.

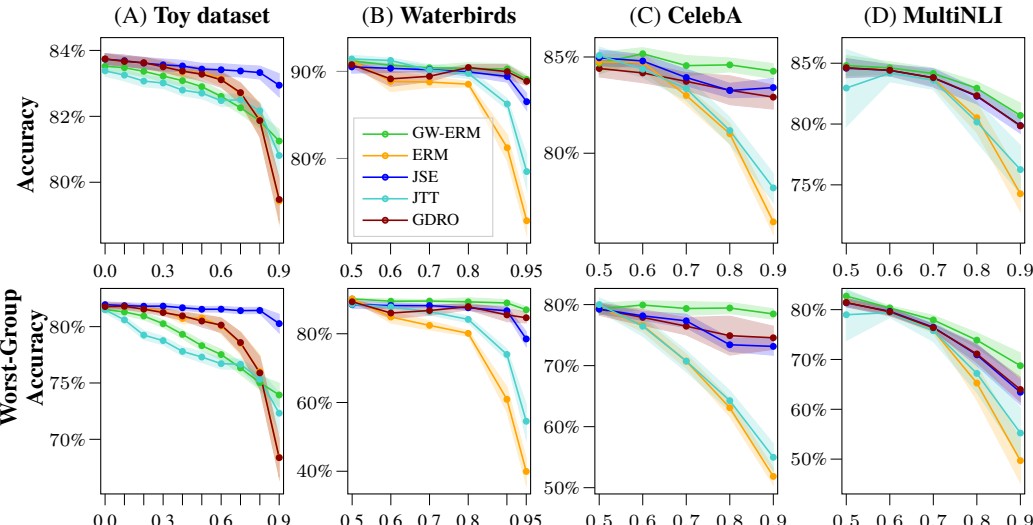

Figure 5: **OOD generalization, compared to instance reweighing algorithms**: same setup as in Figure 3. We added empirical risk minimization (ERM) as the default baseline.

## 5 Conclusion and discussion

This paper has introduced and empirically tested joint subspace estimation (JSE), a novel post-hoc concept-removal method that improves the interpretability and control of last-layer neural representations in the presence of spurious correlations. JSE consistently outperforms existing concept-removal methods, despite making particular assumptions (linearity, orthogonality) about the structure of the embedding space. Future work should consider developing tests for these assumptions, or see whether they can be relaxed. One example is to see if we can jointly estimate subspaces based on their non-linear relationship with the spurious and main-task labels, as done by Ravfogel et al. (2022b) for RLACE.

JSE is also competitive compared to other state-of-the-art techniques specifically aimed at OOD generalization in the presence of spurious correlations. We emphasize that concept-removal methods such as JSE have the added benefit of increasing the interpretability of the neural network.

Our results also highlight the difficulty of separating different concepts in the [CLS] embeddings of BERT. All post-hoc concept-removal methods we consider perform relatively better for the vision datasets than the MultiNLI dataset. This highlights the need to better disentangle concepts in embeddings of large language models, for instance through different training procedures, similar to the work of Zhang et al. (2021).

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

APPENDIX OUTLINE

This appendix is organized as follows. In Section A, we give more details on the results for the four datasets (Waterbirds, CelebA, MultiNLI and Toy) used in the experiments. Section B contains several additional results for the Toy dataset. In Section C we present additional details on the implementation of the JSE algorithm. In Section D we lay out the testing procedure for the JSE algorithm. In Section E, we compare JSE to other instance reweighing methods in the case where limited spurious concept labels are available. Finally, Section F provides a more detailed description of the datasets, as well as implementation details for the experiments.

# A  FULL SET OF RESULTS FOR WATERBIRDS, CELEBA, MULTINLI AND TOY DATASET

| Method | Accuracy | $p_{\mathrm{train}}(y_{\mathrm{mt}} = y\|y_{\mathrm{sp}} = y)$ | | | | |
|---|---|---|---|---|---|---|
| | | 0.5 | 0.6 | 0.7 | 0.8 | 0.9 |
| JSE | $y_{\mathrm{mt}} = 0, y_{\mathrm{sp}} = 0$ | 91.88 (0.45) | 91.65 (0.30) | 91.20 (0.16) | 91.13 (0.41) | 90.23 (0.79) |
| | $y_{\mathrm{mt}} = 0, y_{\mathrm{sp}} = 1$ | 88.59 (0.60) | 88.22 (0.41) | 88.19 (0.21) | 87.56 (0.39) | 87.34 (0.49) |
| | $y_{\mathrm{mt}} = 1, y_{\mathrm{sp}} = 0$ | 90.98 (0.22) | 91.35 (0.18) | 91.81 (0.10) | 91.64 (0.24) | 91.81 (0.24) |
| | $y_{\mathrm{mt}} = 1, y_{\mathrm{sp}} = 1$ | 91.99 (0.18) | 92.28 (0.14) | 92.06 (0.13) | 92.38 (0.19) | 91.48 (0.22) |
| | Worst-group | 88.48 (0.58) | 88.21 (0.41) | 88.19 (0.21) | 87.56 (0.39) | 86.70 (0.62) |
| | Average | 90.51 (0.37) | 90.35 (0.24) | 90.19 (0.11) | 89.93 (0.23) | 89.42 (0.34) |
| ERM | $y_{\mathrm{mt}} = 0, y_{\mathrm{sp}} = 0$ | 91.70 (0.19) | 90.88 (0.82) | 93.14 (0.29) | 95.27 (0.06) | 96.17 (0.25) |
| | $y_{\mathrm{mt}} = 0, y_{\mathrm{sp}} = 1$ | 91.10 (0.20) | 84.87 (0.90) | 82.43 (0.66) | 80.12 (0.41) | 60.92 (1.85) |
| | $y_{\mathrm{mt}} = 1, y_{\mathrm{sp}} = 0$ | 91.18 (0.15) | 91.32 (0.36) | 90.49 (0.21) | 88.32 (0.11) | 84.72 (0.43) |
| | $y_{\mathrm{mt}} = 1, y_{\mathrm{sp}} = 1$ | 90.40 (0.14) | 93.22 (0.18) | 94.03 (0.15) | 94.59 (0.09) | 96.56 (0.16) |
| | Worst-group | 90.13 (0.10) | 84.87 (0.90) | 82.43 (0.66) | 80.12 (0.41) | 60.92 (1.85) |
| | Average | 91.27 (0.11) | 88.85 (0.60) | 88.78 (0.32) | 88.53 (0.16) | 81.23 (0.76) |
| INLP | $y_{\mathrm{mt}} = 0, y_{\mathrm{sp}} = 0$ | 93.89 (0.13) | 90.13 (0.78) | 88.76 (0.61) | 85.70 (0.58) | 82.38 (0.50) |
| | $y_{\mathrm{mt}} = 0, y_{\mathrm{sp}} = 1$ | 87.14 (0.30) | 86.35 (0.97) | 88.54 (0.63) | 91.90 (0.36) | 91.83 (0.24) |
| | $y_{\mathrm{mt}} = 1, y_{\mathrm{sp}} = 0$ | 89.41 (0.20) | 91.71 (0.30) | 92.64 (0.17) | 93.23 (0.21) | 92.73 (0.26) |
| | $y_{\mathrm{mt}} = 1, y_{\mathrm{sp}} = 1$ | 92.38 (0.13) | 92.21 (0.26) | 91.08 (0.29) | 89.08 (0.26) | 85.13 (0.37) |
| | Worst-group | 87.07 (0.28) | 86.33 (0.96) | 88.04 (0.58) | 85.70 (0.58) | 82.26 (0.45) |
| | Average | 90.60 (0.12) | 89.07 (0.62) | 89.36 (0.42) | 89.32 (0.31) | 87.51 (0.21) |
| RLACE | $y_{\mathrm{mt}} = 0, y_{\mathrm{sp}} = 0$ | 92.21 (0.19) | 86.27 (0.96) | 81.18 (0.90) | 77.06 (0.69) | 68.14 (0.77) |
| | $y_{\mathrm{mt}} = 0, y_{\mathrm{sp}} = 1$ | 90.57 (0.25) | 91.44 (0.60) | 94.26 (0.36) | 96.27 (0.13) | 96.45 (0.12) |
| | $y_{\mathrm{mt}} = 1, y_{\mathrm{sp}} = 0$ | 90.83 (0.16) | 93.64 (0.22) | 94.89 (0.15) | 95.40 (0.17) | 95.26 (0.17) |
| | $y_{\mathrm{mt}} = 1, y_{\mathrm{sp}} = 1$ | 90.85 (0.19) | 89.52 (0.35) | 86.95 (0.33) | 81.78 (0.37) | 70.06 (0.61) |
| | Worst-group | 89.88 (0.15) | 85.94 (0.89) | 81.18 (0.90) | 77.06 (0.69) | 67.33 (0.71) |
| | Average | 91.27 (0.12) | 89.46 (0.54) | 88.43 (0.43) | 87.09 (0.29) | 82.37 (0.29) |
| ADV | $y_{\mathrm{mt}} = 0, y_{\mathrm{sp}} = 0$ | 91.66 (0.25) | 93.54 (0.34) | 94.50 (0.33) | 96.41 (0.19) | 97.19 (0.21) |
| | $y_{\mathrm{mt}} = 0, y_{\mathrm{sp}} = 1$ | 90.71 (0.30) | 86.00 (0.61) | 82.41 (0.60) | 76.52 (0.81) | 64.04 (1.47) |
| | $y_{\mathrm{mt}} = 1, y_{\mathrm{sp}} = 0$ | 88.77 (0.38) | 87.23 (0.48) | 84.59 (0.58) | 79.91 (0.64) | 70.76 (1.21) |
| | $y_{\mathrm{mt}} = 1, y_{\mathrm{sp}} = 1$ | 87.57 (0.33) | 90.37 (0.29) | 91.74 (0.26) | 92.53 (0.27) | 94.05 (0.22) |
| | Worst-group | 87.25 (0.31) | 84.97 (0.48) | 81.97 (0.51) | 75.95 (0.69) | 63.68 (1.34) |
| | Average | 89.67 (0.20) | 89.32 (0.19) | 88.36 (0.22) | 86.44 (0.27) | 81.70 (0.56) |

Table 1: **Results for the Waterbirds dataset**: Table shows the average, worst-group, and per-group accuracy on a test set where $p_{\mathrm{OOD}}(y_{\mathrm{mt}} = y|y_{\mathrm{sp}} = y) = 0.5$, with $y \in \{0, 1\}$, as a function of $p_{\mathrm{train}}(y_{\mathrm{mt}} = y|y_{\mathrm{sp}} = y)$. Each accuracy is obtained by averaging over 20 runs. Standard error is reported between brackets.

| Method | Accuracy | $p_{\text{train}}(y_{\text{mt}} = y \mid y_{\text{sp}} = y)$ | | | | |
|---|---|---|---|---|---|---|
| | | 0.5 | 0.6 | 0.7 | 0.8 | 0.9 |
| JSE | $y_{\text{mt}} = 0, y_{\text{sp}} = 0$ | 85.83 (0.37) | 87.17 (0.33) | 87.32 (0.59) | 88.32 (0.49) | 89.41 (0.60) |
| | $y_{\text{mt}} = 0, y_{\text{sp}} = 1$ | 79.36 (0.27) | 79.15 (0.29) | 78.03 (0.59) | 77.48 (0.52) | 78.86 (0.66) |
| | $y_{\text{mt}} = 1, y_{\text{sp}} = 0$ | 81.94 (0.53) | 80.35 (0.79) | 76.48 (0.81) | 75.27 (0.85) | 73.81 (1.22) |
| | $y_{\text{mt}} = 1, y_{\text{sp}} = 1$ | 91.84 (0.33) | 91.80 (0.38) | 91.51 (0.49) | 91.71 (0.55) | 91.96 (0.44) |
| | Worst-group | 79.25 (0.27) | 78.29 (0.52) | 75.87 (0.70) | 74.66 (0.74) | 73.69 (1.20) |
| | Average | 84.74 (0.20) | 84.62 (0.22) | 83.33 (0.23) | 83.19 (0.18) | 83.51 (0.25) |
| ERM | $y_{\text{mt}} = 0, y_{\text{sp}} = 0$ | 85.16 (0.52) | 89.46 (0.41) | 91.92 (0.28) | 94.24 (0.18) | 95.92 (0.23) |
| | $y_{\text{mt}} = 0, y_{\text{sp}} = 1$ | 79.79 (0.38) | 77.60 (0.36) | 74.36 (0.56) | 70.60 (0.63) | 61.10 (0.85) |
| | $y_{\text{mt}} = 1, y_{\text{sp}} = 0$ | 82.30 (0.59) | 78.23 (0.76) | 71.11 (0.67) | 63.10 (0.74) | 51.85 (0.69) |
| | $y_{\text{mt}} = 1, y_{\text{sp}} = 1$ | 91.72 (0.40) | 93.59 (0.27) | 94.61 (0.29) | 96.11 (0.19) | 96.86 (0.12) |
| | Worst-group | 79.57 (0.34) | 76.62 (0.54) | 70.73 (0.59) | 63.10 (0.74) | 51.85 (0.69) |
| | Average | 84.74 (0.19) | 84.72 (0.22) | 83.00 (0.24) | 81.01 (0.27) | 76.43 (0.31) |
| INLP | $y_{\text{mt}} = 0, y_{\text{sp}} = 0$ | 81.44 (0.40) | 75.40 (0.46) | 67.43 (0.51) | 57.94 (0.47) | 52.55 (0.60) |
| | $y_{\text{mt}} = 0, y_{\text{sp}} = 1$ | 81.40 (0.26) | 84.78 (0.23) | 86.64 (0.49) | 87.39 (0.41) | 87.49 (0.35) |
| | $y_{\text{mt}} = 1, y_{\text{sp}} = 0$ | 84.78 (0.56) | 88.04 (0.48) | 89.27 (0.47) | 88.09 (0.46) | 84.94 (0.65) |
| | $y_{\text{mt}} = 1, y_{\text{sp}} = 1$ | 88.86 (0.35) | 82.87 (0.57) | 75.21 (0.49) | 67.20 (0.58) | 58.36 (0.57) |
| | Worst-group | 80.68 (0.28) | 75.40 (0.46) | 67.43 (0.51) | 57.94 (0.47) | 52.36 (0.57) |
| | Average | 84.12 (0.22) | 82.77 (0.24) | 79.64 (0.25) | 75.16 (0.25) | 70.84 (0.29) |
| RLACE | $y_{\text{mt}} = 0, y_{\text{sp}} = 0$ | 83.14 (0.46) | 77.50 (0.57) | 70.82 (0.54) | 62.25 (0.59) | 57.63 (0.96) |
| | $y_{\text{mt}} = 0, y_{\text{sp}} = 1$ | 81.10 (0.36) | 84.29 (0.27) | 86.44 (0.39) | 86.88 (0.35) | 85.87 (0.56) |
| | $y_{\text{mt}} = 1, y_{\text{sp}} = 0$ | 84.64 (0.49) | 88.46 (0.54) | 89.96 (0.46) | 90.29 (0.43) | 85.10 (1.03) |
| | $y_{\text{mt}} = 1, y_{\text{sp}} = 1$ | 90.39 (0.42) | 84.44 (0.55) | 76.68 (0.50) | 69.23 (0.76) | 62.94 (1.04) |
| | Worst-group | 80.91 (0.33) | 77.50 (0.57) | 70.82 (0.54) | 62.25 (0.59) | 57.35 (0.97) |
| | Average | 84.82 (0.23) | 83.67 (0.27) | 80.97 (0.25) | 77.16 (0.33) | 72.88 (0.29) |
| ADV | $y_{\text{mt}} = 0, y_{\text{sp}} = 0$ | 86.31 (0.38) | 89.47 (0.30) | 92.17 (0.34) | 93.64 (0.28) | 95.91 (0.18) |
| | $y_{\text{mt}} = 0, y_{\text{sp}} = 1$ | 80.49 (0.42) | 77.79 (0.45) | 74.47 (0.57) | 70.04 (0.68) | 59.17 (1.10) |
| | $y_{\text{mt}} = 1, y_{\text{sp}} = 0$ | 81.08 (0.91) | 75.28 (0.61) | 69.97 (0.80) | 62.71 (0.63) | 50.26 (0.71) |
| | $y_{\text{mt}} = 1, y_{\text{sp}} = 1$ | 91.86 (0.36) | 93.87 (0.36) | 95.08 (0.26) | 96.21 (0.21) | 96.84 (0.23) |
| | Worst-group | 78.89 (0.60) | 74.97 (0.57) | 69.92 (0.78) | 62.71 (0.63) | 50.08 (0.69) |
| | Average | 84.93 (0.24) | 84.10 (0.22) | 82.92 (0.26) | 80.65 (0.28) | 75.55 (0.42) |

Table 2: **Results for the celebA dataset**: Table shows the average, worst-group, and per-group accuracy on a test set where $p_{\text{OOD}}(y_{\text{mt}} = y \mid y_{\text{sp}} = y) = 0.5$, with $y \in \{0, 1\}$, as a function of $p_{\text{train}}(y_{\text{mt}} = y \mid y_{\text{sp}} = y)$. Each accuracy is obtained by averaging over 20 runs. Standard error is reported between brackets.

| Method | Accuracy | $p_{\text{train}}(y_{\text{mt}} = y \mid y_{\text{sp}} = y)$ | | | | |
|---|---|---|---|---|---|---|
| | | 0.5 | 0.6 | 0.7 | 0.8 | 0.9 |
| JSE | $y_{\text{mt}} = 0, y_{\text{sp}} = 0$ | 87.49 (0.69) | 88.83 (0.52) | 90.34 (0.36) | 92.53 (0.47) | 93.94 (0.56) |
| | $y_{\text{mt}} = 0, y_{\text{sp}} = 1$ | 87.31 (0.42) | 84.30 (0.66) | 82.24 (0.98) | 78.21 (0.92) | 73.97 (1.22) |
| | $y_{\text{mt}} = 1, y_{\text{sp}} = 0$ | 81.74 (0.54) | 79.66 (0.35) | 76.46 (0.35) | 70.93 (1.10) | 63.42 (1.26) |
| | $y_{\text{mt}} = 1, y_{\text{sp}} = 1$ | 81.81 (0.31) | 84.86 (0.46) | 86.14 (0.86) | 87.47 (0.42) | 88.18 (0.96) |
| | Worst-group | 81.38 (0.41) | 79.66 (0.35) | 76.46 (0.35) | 70.93 (1.10) | 63.42 (1.26) |
| | Average | 84.59 (0.40) | 84.42 (0.23) | 83.80 (0.38) | 82.28 (0.32) | 79.88 (0.33) |
| ERM | $y_{\text{mt}} = 0, y_{\text{sp}} = 0$ | 87.42 (0.69) | 88.99 (0.55) | 91.06 (0.37) | 94.30 (0.43) | 97.06 (0.29) |
| | $y_{\text{mt}} = 0, y_{\text{sp}} = 1$ | 87.20 (0.49) | 83.73 (0.60) | 80.48 (1.11) | 70.24 (1.91) | 51.84 (3.09) |
| | $y_{\text{mt}} = 1, y_{\text{sp}} = 0$ | 81.87 (0.48) | 79.65 (0.34) | 75.84 (0.27) | 65.92 (1.84) | 52.05 (1.37) |
| | $y_{\text{mt}} = 1, y_{\text{sp}} = 1$ | 81.92 (0.31) | 85.39 (0.55) | 87.57 (0.59) | 91.60 (0.66) | 96.13 (0.70) |
| | Worst-group | 81.50 (0.35) | 79.65 (0.34) | 75.84 (0.27) | 65.26 (1.72) | 49.70 (2.22) |
| | Average | 84.60 (0.39) | 84.44 (0.24) | 83.74 (0.31) | 80.52 (0.62) | 74.27 (0.73) |
| INLP | $y_{\text{mt}} = 0, y_{\text{sp}} = 0$ | 87.44 (0.69) | 88.96 (0.57) | 90.50 (0.41) | 75.09 (4.41) | 56.45 (1.77) |
| | $y_{\text{mt}} = 0, y_{\text{sp}} = 1$ | 87.20 (0.49) | 83.84 (0.63) | 81.95 (0.77) | 65.95 (4.04) | 58.26 (4.10) |
| | $y_{\text{mt}} = 1, y_{\text{sp}} = 0$ | 81.86 (0.49) | 79.73 (0.33) | 76.51 (0.34) | 62.10 (4.29) | 50.64 (3.92) |
| | $y_{\text{mt}} = 1, y_{\text{sp}} = 1$ | 81.94 (0.32) | 85.30 (0.52) | 86.43 (0.67) | 63.12 (7.22) | 54.54 (3.12) |
| | Worst-group | 81.47 (0.36) | 79.73 (0.33) | 76.51 (0.34) | 57.62 (4.97) | 48.13 (3.12) |
| | Average | 84.61 (0.39) | 84.46 (0.22) | 83.85 (0.36) | 66.56 (4.51) | 54.97 (2.06) |
| RLACE | $y_{\text{mt}} = 0, y_{\text{sp}} = 0$ | 87.16 (0.72) | 88.90 (0.47) | 81.89 (0.82) | 76.61 (1.60) | 69.33 (1.95) |
| | $y_{\text{mt}} = 0, y_{\text{sp}} = 1$ | 87.24 (0.55) | 84.02 (0.72) | 91.09 (0.39) | 92.02 (0.69) | 93.70 (0.38) |
| | $y_{\text{mt}} = 1, y_{\text{sp}} = 0$ | 81.64 (0.51) | 79.81 (0.31) | 84.59 (1.12) | 86.27 (0.87) | 88.90 (1.04) |
| | $y_{\text{mt}} = 1, y_{\text{sp}} = 1$ | 82.16 (0.32) | 85.01 (0.43) | 72.16 (1.18) | 66.69 (1.51) | 61.15 (1.20) |
| | Worst-group | 81.52 (0.44) | 79.81 (0.31) | 72.16 (1.18) | 66.69 (1.51) | 61.15 (1.20) |
| | Average | 84.55 (0.43) | 84.43 (0.25) | 82.43 (0.35) | 80.40 (0.64) | 78.27 (0.41) |

Table 3: **Results for the MultiNLI dataset**: Table shows the average, worst-group, and per-group accuracy on a test set where $p_{\text{OOD}}(y_{\text{mt}} = y \mid y_{\text{sp}} = y) = 0.5$, with $y \in \{0, 1\}$, as a function of $p_{\text{train}}(y_{\text{mt}} = y \mid y_{\text{sp}} = y)$. Each accuracy is obtained by averaging over 5 runs. Standard error is reported between brackets.

| Method | Accuracy | $\rho$ 0.0 | 0.1 | 0.2 | 0.3 | 0.4 |
|---|---|---|---|---|---|---|
| JSE | $y_{\mathrm{mt}}=0, y_{\mathrm{sp}}=0$ | 83.71 (0.17) | 83.62 (0.18) | 83.46 (0.18) | 83.36 (0.18) | 83.40 (0.16) |
| | $y_{\mathrm{mt}}=0, y_{\mathrm{sp}}=1$ | 83.67 (0.16) | 83.56 (0.17) | 83.36 (0.19) | 83.31 (0.18) | 83.24 (0.19) |
| | $y_{\mathrm{mt}}=1, y_{\mathrm{sp}}=0$ | 83.73 (0.19) | 83.71 (0.20) | 83.79 (0.19) | 83.71 (0.18) | 83.63 (0.20) |
| | $y_{\mathrm{mt}}=1, y_{\mathrm{sp}}=1$ | 83.77 (0.15) | 83.83 (0.16) | 83.81 (0.16) | 83.84 (0.17) | 83.82 (0.17) |
| | Worst-group | 81.95 (0.12) | 81.86 (0.13) | 81.81 (0.13) | 81.82 (0.13) | 81.68 (0.12) |
| | Average | 83.73 (0.09) | 83.69 (0.09) | 83.61 (0.09) | 83.56 (0.09) | 83.53 (0.09) |
| ERM | $y_{\mathrm{mt}}=0, y_{\mathrm{sp}}=0$ | 83.55 (0.16) | 83.46 (0.18) | 83.42 (0.19) | 83.52 (0.21) | 83.85 (0.23) |
| | $y_{\mathrm{mt}}=0, y_{\mathrm{sp}}=1$ | 83.82 (0.17) | 83.73 (0.17) | 83.38 (0.19) | 83.17 (0.22) | 82.44 (0.29) |
| | $y_{\mathrm{mt}}=1, y_{\mathrm{sp}}=0$ | 83.89 (0.19) | 83.83 (0.20) | 83.72 (0.20) | 83.50 (0.21) | 82.82 (0.35) |
| | $y_{\mathrm{mt}}=1, y_{\mathrm{sp}}=1$ | 83.61 (0.16) | 83.72 (0.16) | 83.83 (0.16) | 83.92 (0.20) | 84.22 (0.21) |
| | Worst-group | 81.90 (0.13) | 81.80 (0.13) | 81.63 (0.13) | 81.41 (0.15) | 80.72 (0.25) |
| | Average | 83.74 (0.08) | 83.70 (0.08) | 83.60 (0.09) | 83.54 (0.09) | 83.35 (0.10) |
| INLP | $y_{\mathrm{mt}}=0, y_{\mathrm{sp}}=0$ | 83.76 (0.18) | 83.07 (0.29) | 78.93 (0.80) | 72.41 (1.16) | 64.74 (1.17) |
| | $y_{\mathrm{mt}}=0, y_{\mathrm{sp}}=1$ | 83.64 (0.17) | 83.44 (0.24) | 80.08 (0.74) | 74.74 (1.10) | 67.58 (1.24) |
| | $y_{\mathrm{mt}}=1, y_{\mathrm{sp}}=0$ | 83.60 (0.22) | 83.66 (0.25) | 80.52 (0.73) | 75.42 (1.08) | 68.16 (1.21) |
| | $y_{\mathrm{mt}}=1, y_{\mathrm{sp}}=1$ | 83.79 (0.16) | 83.43 (0.23) | 79.52 (0.82) | 72.99 (1.14) | 65.16 (1.16) |
| | Worst-group | 81.72 (0.14) | 81.26 (0.24) | 77.32 (0.81) | 70.58 (1.19) | 61.41 (1.24) |
| | Average | 83.70 (0.09) | 83.41 (0.18) | 79.77 (0.75) | 73.90 (1.09) | 66.45 (1.11) |
| RLACE | $y_{\mathrm{mt}}=0, y_{\mathrm{sp}}=0$ | 83.69 (0.21) | 81.44 (0.23) | 78.52 (0.26) | 75.37 (0.26) | 72.31 (0.27) |
| | $y_{\mathrm{mt}}=0, y_{\mathrm{sp}}=1$ | 83.49 (0.20) | 85.42 (0.18) | 86.90 (0.17) | 88.25 (0.14) | 89.08 (0.13) |
| | $y_{\mathrm{mt}}=1, y_{\mathrm{sp}}=0$ | 83.57 (0.25) | 85.43 (0.23) | 87.13 (0.18) | 88.47 (0.16) | 89.42 (0.14) |
| | $y_{\mathrm{mt}}=1, y_{\mathrm{sp}}=1$ | 83.60 (0.21) | 81.67 (0.25) | 79.27 (0.26) | 75.98 (0.28) | 72.99 (0.28) |
| | Worst-group | 81.20 (0.16) | 80.30 (0.21) | 77.72 (0.24) | 74.53 (0.25) | 71.38 (0.25) |
| | Average | 83.59 (0.09) | 83.49 (0.09) | 82.96 (0.10) | 82.02 (0.11) | 80.95 (0.11) |

Table 4: **Results for the Toy dataset for** $\rho \in \{0.0, 0.1, 0.2, 0.3, 0.4\}$. Table shows the average, worst-group, and per-group accuracy on a test set without spurious correlation, as a function of the spurious correlation in the training data. Each accuracy is obtained by averaging over 100 runs. Standard error is reported between brackets.

| Method | Accuracy | $\rho$ 0.5 | 0.6 | 0.7 | 0.8 | 0.9 |
|---|---|---|---|---|---|---|
| JSE | $y_{\mathrm{mt}}=0, y_{\mathrm{sp}}=0$ | 83.27 (0.16) | 83.31 (0.17) | 83.25 (0.18) | 83.30 (0.18) | 83.49 (0.26) |
| | $y_{\mathrm{mt}}=0, y_{\mathrm{sp}}=1$ | 83.20 (0.18) | 83.06 (0.18) | 82.99 (0.20) | 83.06 (0.21) | 81.87 (0.45) |
| | $y_{\mathrm{mt}}=1, y_{\mathrm{sp}}=0$ | 83.57 (0.20) | 83.54 (0.21) | 83.48 (0.22) | 83.31 (0.20) | 82.35 (0.47) |
| | $y_{\mathrm{mt}}=1, y_{\mathrm{sp}}=1$ | 83.67 (0.18) | 83.71 (0.18) | 83.76 (0.18) | 83.65 (0.19) | 84.05 (0.25) |
| | Worst-group | 81.54 (0.12) | 81.55 (0.13) | 81.42 (0.14) | 81.44 (0.14) | 80.27 (0.42) |
| | Average | 83.43 (0.09) | 83.41 (0.10) | 83.38 (0.09) | 83.33 (0.10) | 82.94 (0.18) |
| ERM | $y_{\mathrm{mt}}=0, y_{\mathrm{sp}}=0$ | 83.61 (0.22) | 83.76 (0.27) | 84.60 (0.31) | 86.00 (0.31) | 88.72 (0.28) |
| | $y_{\mathrm{mt}}=0, y_{\mathrm{sp}}=1$ | 82.76 (0.28) | 82.10 (0.40) | 80.39 (0.53) | 77.80 (0.77) | 69.97 (0.88) |
| | $y_{\mathrm{mt}}=1, y_{\mathrm{sp}}=0$ | 83.03 (0.30) | 82.50 (0.42) | 80.62 (0.60) | 77.78 (0.79) | 70.11 (0.94) |
| | $y_{\mathrm{mt}}=1, y_{\mathrm{sp}}=1$ | 83.93 (0.22) | 84.16 (0.24) | 84.97 (0.28) | 85.99 (0.32) | 88.86 (0.26) |
| | Worst-group | 80.76 (0.22) | 80.06 (0.36) | 78.51 (0.53) | 76.01 (0.77) | 68.41 (0.91) |
| | Average | 83.35 (0.09) | 83.15 (0.12) | 82.67 (0.17) | 81.90 (0.26) | 79.43 (0.35) |
| INLP | $y_{\mathrm{mt}}=0, y_{\mathrm{sp}}=0$ | 59.94 (1.03) | 58.54 (0.90) | 55.66 (0.81) | 54.71 (0.68) | 53.06 (0.69) |
| | $y_{\mathrm{mt}}=0, y_{\mathrm{sp}}=1$ | 62.83 (1.19) | 61.77 (1.26) | 58.31 (1.29) | 56.64 (1.27) | 56.01 (1.53) |
| | $y_{\mathrm{mt}}=1, y_{\mathrm{sp}}=0$ | 62.82 (1.19) | 60.43 (1.31) | 58.98 (1.26) | 57.09 (1.26) | 58.23 (1.48) |
| | $y_{\mathrm{mt}}=1, y_{\mathrm{sp}}=1$ | 60.52 (1.01) | 57.87 (0.95) | 56.60 (0.82) | 55.09 (0.65) | 55.14 (0.70) |
| | Worst-group | 55.51 (1.06) | 53.06 (0.95) | 50.06 (0.77) | 48.59 (0.65) | 47.06 (0.56) |
| | Average | 61.58 (0.94) | 59.71 (0.89) | 57.44 (0.76) | 55.93 (0.68) | 55.67 (0.74) |
| RLACE | $y_{\mathrm{mt}}=0, y_{\mathrm{sp}}=0$ | 68.59 (0.27) | 64.77 (0.27) | 60.84 (0.29) | 57.30 (0.28) | 53.89 (0.30) |
| | $y_{\mathrm{mt}}=0, y_{\mathrm{sp}}=1$ | 89.97 (0.14) | 90.60 (0.13) | 91.02 (0.12) | 91.21 (0.13) | 90.71 (0.30) |
| | $y_{\mathrm{mt}}=1, y_{\mathrm{sp}}=0$ | 90.22 (0.13) | 90.87 (0.13) | 91.30 (0.13) | 91.65 (0.14) | 90.89 (0.39) |
| | $y_{\mathrm{mt}}=1, y_{\mathrm{sp}}=1$ | 69.32 (0.33) | 65.42 (0.31) | 61.70 (0.31) | 57.85 (0.32) | 54.39 (0.35) |
| | Worst-group | 67.65 (0.27) | 63.87 (0.27) | 59.74 (0.27) | 55.89 (0.25) | 52.31 (0.27) |
| | Average | 79.53 (0.13) | 77.92 (0.13) | 76.22 (0.13) | 74.50 (0.13) | 72.48 (0.20) |

Table 5: **Results for the Toy dataset for** $\rho \in \{0.5, 0.6, 0.7, 0.8, 0.9\}$. Table shows the average, worst-group, and per-group accuracy on a test set without spurious correlation, as a function of the spurious correlation in the training data. Each accuracy is obtained by averaging over 100 runs. Standard error is reported between brackets.

| | | $p_{\text{train}}(y_{\text{mt}} = y | y_{\text{sp}} = y)$ | | | | |
|---|---|---|---|---|---|---|
| **Method** | **Accuracy** | 0.5 | 0.6 | 0.7 | 0.8 | 0.9 |
| GW-ERM | $y_{\text{mt}} = 0, y_{\text{sp}} = 0$ | 91.50 (0.19) | 91.21 (0.33) | 90.34 (0.26) | 89.53 (0.62) | 90.12 (0.60) |
| | $y_{\text{mt}} = 0, y_{\text{sp}} = 1$ | 91.07 (0.21) | 89.67 (0.38) | 89.82 (0.21) | 90.46 (0.35) | 89.75 (0.33) |
| | $y_{\text{mt}} = 1, y_{\text{sp}} = 0$ | 91.36 (0.16) | 91.46 (0.21) | 92.18 (0.10) | 91.81 (0.19) | 91.83 (0.25) |
| | $y_{\text{mt}} = 1, y_{\text{sp}} = 1$ | 90.48 (0.15) | 91.71 (0.17) | 90.98 (0.19) | 91.26 (0.18) | 91.11 (0.15) |
| | Worst-group | 91.20 (0.12) | 90.70 (0.23) | 90.41 (0.13) | 90.34 (0.34) | 90.27 (0.30) |
| | Average | 91.20 (0.12) | 90.70 (0.23) | 90.41 (0.13) | 90.34 (0.34) | 90.27 (0.30) |
| GDRO | $y_{\text{mt}} = 0, y_{\text{sp}} = 0$ | 91.09 (0.36) | 90.48 (0.59) | 90.19 (0.56) | 92.04 (0.43) | 93.17 (0.58) |
| | $y_{\text{mt}} = 0, y_{\text{sp}} = 1$ | 90.48 (0.33) | 86.27 (0.85) | 87.24 (0.73) | 88.48 (0.60) | 86.43 (1.16) |
| | $y_{\text{mt}} = 1, y_{\text{sp}} = 0$ | 90.97 (0.24) | 91.29 (0.33) | 91.92 (0.23) | 90.43 (0.28) | 89.53 (0.40) |
| | $y_{\text{mt}} = 1, y_{\text{sp}} = 1$ | 90.46 (0.24) | 92.37 (0.30) | 92.03 (0.30) | 91.72 (0.29) | 91.60 (0.47) |
| | Worst-group | 89.31 (0.21) | 86.03 (0.81) | 86.76 (0.72) | 87.88 (0.47) | 85.49 (0.99) |
| | Average | 90.77 (0.15) | 89.14 (0.44) | 89.43 (0.30) | 90.44 (0.18) | 89.97 (0.41) |
| JTT | $y_{\text{mt}} = 0, y_{\text{sp}} = 0$ | 91.73 (0.34) | 94.11 (0.24) | 93.57 (0.46) | 95.15 (0.33) | 96.76 (0.18) |
| | $y_{\text{mt}} = 0, y_{\text{sp}} = 1$ | 92.10 (0.42) | 89.31 (0.29) | 86.56 (0.49) | 84.37 (0.52) | 73.98 (0.81) |
| | $y_{\text{mt}} = 1, y_{\text{sp}} = 0$ | 90.29 (0.22) | 88.31 (0.25) | 88.67 (0.30) | 86.40 (0.35) | 83.83 (0.48) |
| | $y_{\text{mt}} = 1, y_{\text{sp}} = 1$ | 89.14 (0.25) | 91.03 (0.22) | 92.33 (0.17) | 93.25 (0.12) | 94.93 (0.16) |
| | Worst-group | 88.57 (0.25) | 87.79 (0.15) | 86.44 (0.48) | 84.18 (0.51) | 73.98 (0.81) |
| | Average | 91.42 (0.22) | 91.26 (0.14) | 90.16 (0.31) | 89.77 (0.29) | 86.26 (0.32) |

Table 6: **Results of instance reweighing methods for the Waterbirds dataset**: Table shows the average, worst-group, and per-group accuracy on a test set where $p_{\text{OOD}}(y_{\text{mt}} = y | y_{\text{sp}} = y) = 0.5$, with $y \in \{0, 1\}$, as a function of $p_{\text{train}}(y_{\text{mt}} = y | y_{\text{sp}} = y)$. Each accuracy is obtained by averaging over 20 runs. Standard error is reported between brackets.

| | | $p_{\text{train}}(y_{\text{mt}} = y | y_{\text{sp}} = y)$ | | | | |
|---|---|---|---|---|---|---|
| **Method** | **Accuracy** | 0.5 | 0.6 | 0.7 | 0.8 | 0.9 |
| GW-ERM | $y_{\text{mt}} = 0, y_{\text{sp}} = 0$ | 85.10 (0.39) | 85.83 (0.53) | 85.94 (0.39) | 85.55 (0.46) | 85.91 (0.52) |
| | $y_{\text{mt}} = 0, y_{\text{sp}} = 1$ | 79.50 (0.51) | 80.16 (0.34) | 79.87 (0.60) | 80.33 (0.41) | 80.34 (0.60) |
| | $y_{\text{mt}} = 1, y_{\text{sp}} = 0$ | 82.34 (0.64) | 82.94 (0.57) | 80.85 (0.54) | 80.81 (0.53) | 79.86 (0.57) |
| | $y_{\text{mt}} = 1, y_{\text{sp}} = 1$ | 91.76 (0.44) | 91.74 (0.31) | 91.55 (0.36) | 91.67 (0.30) | 90.94 (0.25) |
| | Worst-group | 79.29 (0.47) | 79.93 (0.36) | 79.36 (0.57) | 79.44 (0.46) | 78.47 (0.43) |
| | Average | 84.68 (0.29) | 85.17 (0.16) | 84.55 (0.27) | 84.59 (0.18) | 84.26 (0.20) |
| GDRO | $y_{\text{mt}} = 0, y_{\text{sp}} = 0$ | 84.55 (0.83) | 86.62 (0.65) | 87.06 (0.74) | 86.80 (0.94) | 86.57 (1.06) |
| | $y_{\text{mt}} = 0, y_{\text{sp}} = 1$ | 80.03 (0.48) | 78.83 (0.44) | 78.60 (0.68) | 78.48 (0.89) | 78.46 (0.95) |
| | $y_{\text{mt}} = 1, y_{\text{sp}} = 0$ | 82.02 (0.67) | 79.34 (0.94) | 77.32 (0.82) | 76.35 (1.86) | 75.71 (1.27) |
| | $y_{\text{mt}} = 1, y_{\text{sp}} = 1$ | 91.01 (0.62) | 91.90 (0.49) | 91.95 (0.46) | 91.46 (0.59) | 90.92 (0.73) |
| | Worst-group | 79.26 (0.52) | 77.90 (0.65) | 76.46 (0.73) | 74.91 (1.63) | 74.56 (0.97) |
| | Average | 84.40 (0.22) | 84.17 (0.26) | 83.73 (0.23) | 83.27 (0.38) | 82.92 (0.32) |
| JTT | $y_{\text{mt}} = 0, y_{\text{sp}} = 0$ | 86.21 (0.82) | 87.84 (0.62) | 91.38 (0.47) | 93.54 (0.30) | 95.68 (0.24) |
| | $y_{\text{mt}} = 0, y_{\text{sp}} = 1$ | 81.90 (0.48) | 80.34 (0.55) | 76.24 (0.90) | 71.05 (0.87) | 65.83 (0.89) |
| | $y_{\text{mt}} = 1, y_{\text{sp}} = 0$ | 81.72 (0.83) | 76.62 (0.85) | 70.85 (0.91) | 64.26 (0.84) | 54.98 (1.14) |
| | $y_{\text{mt}} = 1, y_{\text{sp}} = 1$ | 90.48 (0.65) | 92.66 (0.37) | 94.95 (0.24) | 95.86 (0.22) | 96.30 (0.23) |
| | Worst-group | 80.00 (0.57) | 76.46 (0.83) | 70.74 (0.88) | 64.26 (0.84) | 54.98 (1.14) |
| | Average | 85.08 (0.23) | 84.36 (0.22) | 83.36 (0.31) | 81.18 (0.35) | 78.20 (0.38) |

Table 7: **Results instance reweighing methods for the celebA dataset**: Table shows the average, worst-group, and per-group accuracy on a test set where $p_{\text{OOD}}(y_{\text{mt}} = y | y_{\text{sp}} = y) = 0.5$, with $y \in \{0, 1\}$, as a function of $p_{\text{train}}(y_{\text{mt}} = y | y_{\text{sp}} = y)$. Each accuracy is obtained by averaging over 20 runs. Standard error is reported between brackets.

| Method | Accuracy | $p_{\text{train}}(y_{\text{mt}} = y \mid y_{\text{sp}} = y)$ | | | | |
| | | 0.5 | 0.6 | 0.7 | 0.8 | 0.9 |
|---|---|---|---|---|---|---|
| GW-ERM | $y_{\text{mt}} = 0, y_{\text{sp}} = 0$ | 86.51 (0.50) | 88.69 (0.46) | 90.21 (0.38) | 91.34 (0.40) | 92.54 (0.54) |
| | $y_{\text{mt}} = 0, y_{\text{sp}} = 1$ | 86.05 (0.51) | 83.55 (0.74) | 81.20 (0.64) | 77.87 (1.27) | 71.73 (1.87) |
| | $y_{\text{mt}} = 1, y_{\text{sp}} = 0$ | 83.46 (0.68) | 80.35 (0.21) | 77.94 (0.47) | 73.87 (0.76) | 68.75 (1.31) |
| | $y_{\text{mt}} = 1, y_{\text{sp}} = 1$ | 83.22 (0.43) | 85.87 (0.49) | 87.28 (0.38) | 88.69 (0.78) | 89.79 (0.93) |
| | Worst-group | 82.72 (0.54) | 80.35 (0.21) | 77.94 (0.47) | 73.87 (0.76) | 68.75 (1.31) |
| | Average | 84.81 (0.42) | 84.62 (0.26) | 84.16 (0.22) | 82.94 (0.28) | 80.70 (0.52) |
| GDRO | $y_{\text{mt}} = 0, y_{\text{sp}} = 0$ | 87.47 (0.66) | 89.07 (0.64) | 90.75 (0.29) | 92.62 (0.34) | 94.05 (0.46) |
| | $y_{\text{mt}} = 0, y_{\text{sp}} = 1$ | 87.50 (0.40) | 84.10 (0.70) | 81.17 (0.95) | 77.49 (1.20) | 71.70 (1.17) |
| | $y_{\text{mt}} = 1, y_{\text{sp}} = 0$ | 81.84 (0.54) | 79.55 (0.41) | 76.37 (0.33) | 71.10 (0.90) | 63.97 (1.25) |
| | $y_{\text{mt}} = 1, y_{\text{sp}} = 1$ | 81.71 (0.43) | 84.93 (0.62) | 86.94 (0.82) | 88.11 (0.60) | 89.70 (0.86) |
| | Worst-group | 81.42 (0.44) | 79.55 (0.41) | 76.37 (0.33) | 71.10 (0.90) | 63.97 (1.25) |
| | Average | 84.63 (0.42) | 84.41 (0.28) | 83.81 (0.34) | 82.33 (0.32) | 79.85 (0.41) |
| JTT | $y_{\text{mt}} = 0, y_{\text{sp}} = 0$ | 83.52 (1.64) | 88.29 (0.83) | 90.99 (0.26) | 92.10 (2.08) | 96.11 (0.83) |
| | $y_{\text{mt}} = 0, y_{\text{sp}} = 1$ | 82.42 (2.40) | 81.97 (1.03) | 78.05 (1.33) | 70.02 (2.20) | 57.92 (3.39) |
| | $y_{\text{mt}} = 1, y_{\text{sp}} = 0$ | 82.82 (3.15) | 79.95 (0.48) | 76.00 (1.06) | 68.58 (2.23) | 56.14 (2.28) |
| | $y_{\text{mt}} = 1, y_{\text{sp}} = 1$ | 83.09 (2.65) | 86.54 (0.66) | 89.34 (0.32) | 90.00 (2.81) | 94.85 (0.66) |
| | Worst-group | 78.98 (2.64) | 79.41 (0.50) | 75.78 (1.10) | 67.20 (1.93) | 55.23 (2.55) |
| | Average | 82.96 (1.61) | 84.19 (0.36) | 83.60 (0.38) | 80.17 (0.84) | 76.26 (1.01) |

Table 8: **Results of instance reweighing methods for the MultiNLI dataset**: Table shows the average, worst-group, and per-group accuracy on a test set where $p_{\text{OOD}}(y_{\text{mt}} = y \mid y_{\text{sp}} = y) = 0.5$, with $y \in \{0, 1\}$, as a function of $p_{\text{train}}(y_{\text{mt}} = y \mid y_{\text{sp}} = y)$. Each accuracy is obtained by averaging over 5 runs. Standard error is reported between brackets.

| Method | Accuracy | $\rho$ | | | | |
| | | 0.0 | 0.1 | 0.2 | 0.3 | 0.4 |
|---|---|---|---|---|---|---|
| GW-ERM | For $y_{\text{mt}} = 0$ and $y_{\text{sp}} = 0$ | 83.26 (0.20) | 82.64 (0.19) | 82.15 (0.21) | 81.25 (0.22) | 80.44 (0.25) |
| | For $y_{\text{mt}} = 0$ and $y_{\text{sp}} = 1$ | 83.56 (0.17) | 83.96 (0.19) | 84.55 (0.18) | 84.69 (0.19) | 85.47 (0.18) |
| | For $y_{\text{mt}} = 1$ and $y_{\text{sp}} = 0$ | 83.79 (0.19) | 84.25 (0.20) | 84.56 (0.17) | 85.13 (0.19) | 85.57 (0.20) |
| | For $y_{\text{mt}} = 1$ and $y_{\text{sp}} = 1$ | 83.45 (0.17) | 83.05 (0.18) | 82.21 (0.19) | 81.82 (0.18) | 80.87 (0.21) |
| | Worst-group | 81.52 (0.14) | 81.30 (0.14) | 80.95 (0.17) | 80.26 (0.17) | 79.31 (0.19) |
| | Average | 83.52 (0.09) | 83.48 (0.08) | 83.36 (0.08) | 83.22 (0.09) | 83.08 (0.09) |
| GDRO | For $y_{\text{mt}} = 0$ and $y_{\text{sp}} = 0$ | 83.63 (0.16) | 83.49 (0.15) | 83.57 (0.17) | 83.44 (0.20) | 83.45 (0.21) |
| | For $y_{\text{mt}} = 0$ and $y_{\text{sp}} = 1$ | 83.93 (0.18) | 83.78 (0.18) | 83.81 (0.20) | 83.49 (0.21) | 83.13 (0.25) |
| | For $y_{\text{mt}} = 1$ and $y_{\text{sp}} = 0$ | 83.54 (0.20) | 83.54 (0.20) | 83.42 (0.21) | 83.21 (0.22) | 82.91 (0.26) |
| | For $y_{\text{mt}} = 1$ and $y_{\text{sp}} = 1$ | 83.81 (0.19) | 83.80 (0.17) | 83.66 (0.20) | 83.82 (0.21) | 83.94 (0.22) |
| | Worst-group | 81.78 (0.15) | 81.81 (0.14) | 81.55 (0.14) | 81.25 (0.16) | 80.95 (0.20) |
| | Average | 83.74 (0.08) | 83.66 (0.08) | 83.63 (0.08) | 83.50 (0.08) | 83.37 (0.08) |
| JTT | For $y_{\text{mt}} = 0$ and $y_{\text{sp}} = 0$ | 83.23 (0.18) | 81.81 (0.23) | 80.60 (0.23) | 80.07 (0.24) | 79.59 (0.30) |
| | For $y_{\text{mt}} = 0$ and $y_{\text{sp}} = 1$ | 83.71 (0.17) | 84.83 (0.18) | 85.80 (0.18) | 86.06 (0.19) | 86.22 (0.25) |
| | For $y_{\text{mt}} = 1$ and $y_{\text{sp}} = 0$ | 83.12 (0.18) | 84.34 (0.20) | 85.21 (0.17) | 85.64 (0.20) | 85.77 (0.25) |
| | For $y_{\text{mt}} = 1$ and $y_{\text{sp}} = 1$ | 83.43 (0.19) | 81.99 (0.17) | 80.66 (0.25) | 80.27 (0.25) | 79.60 (0.30) |
| | Worst-group | 81.49 (0.13) | 80.60 (0.16) | 79.25 (0.20) | 78.76 (0.19) | 77.80 (0.20) |
| | Average | 83.38 (0.08) | 83.25 (0.09) | 83.07 (0.09) | 83.01 (0.08) | 82.80 (0.09) |

Table 9: **Results of instance reweighing methods for the Toy dataset for** $\rho \in \{0.0, 0.1, 0.2, 0.3, 0.4\}$. Table shows the average, worst-group, and per-group accuracy on a test set without spurious correlation, as a function of the spurious correlation in the training data. Each accuracy is obtained by averaging over 100 runs. Standard error is reported between brackets.

| Method | Accuracy | $\rho$ 0.5 | 0.6 | 0.7 | 0.8 | 0.9 |
|---|---|---|---|---|---|---|
| GW-ERM | For $y_{\mathrm{mt}}=0$ and $y_{\mathrm{sp}}=0$ | 79.55 (0.30) | 78.88 (0.30) | 78.37 (0.41) | 77.57 (0.46) | 79.78 (0.57) |
| | For $y_{\mathrm{mt}}=0$ and $y_{\mathrm{sp}}=1$ | 85.65 (0.23) | 85.63 (0.27) | 85.73 (0.40) | 85.59 (0.50) | 82.78 (0.78) |
| | For $y_{\mathrm{mt}}=1$ and $y_{\mathrm{sp}}=0$ | 86.05 (0.22) | 86.19 (0.29) | 86.06 (0.35) | 85.99 (0.47) | 82.69 (0.80) |
| | For $y_{\mathrm{mt}}=1$ and $y_{\mathrm{sp}}=1$ | 80.33 (0.24) | 79.72 (0.33) | 78.90 (0.37) | 78.28 (0.45) | 79.68 (0.59) |
| | Worst-group | 78.31 (0.22) | 77.52 (0.24) | 76.33 (0.29) | 75.03 (0.31) | 73.94 (0.53) |
| | Average | 82.90 (0.09) | 82.60 (0.08) | 82.26 (0.10) | 81.86 (0.11) | 81.25 (0.18) |
| GDRO | For $y_{\mathrm{mt}}=0$ and $y_{\mathrm{sp}}=0$ | 83.64 (0.24) | 83.89 (0.23) | 84.62 (0.29) | 85.70 (0.30) | 88.46 (0.29) |
| | For $y_{\mathrm{mt}}=0$ and $y_{\mathrm{sp}}=1$ | 83.01 (0.34) | 82.25 (0.39) | 80.94 (0.50) | 77.92 (0.78) | 69.99 (0.90) |
| | For $y_{\mathrm{mt}}=1$ and $y_{\mathrm{sp}}=0$ | 82.51 (0.33) | 82.07 (0.36) | 80.51 (0.55) | 77.70 (0.76) | 70.26 (0.85) |
| | For $y_{\mathrm{mt}}=1$ and $y_{\mathrm{sp}}=1$ | 83.93 (0.25) | 84.20 (0.28) | 84.74 (0.31) | 86.10 (0.33) | 89.00 (0.28) |
| | Worst-group | 80.50 (0.27) | 80.14 (0.33) | 78.59 (0.47) | 75.90 (0.73) | 68.37 (0.87) |
| | Average | 83.28 (0.10) | 83.11 (0.12) | 82.72 (0.15) | 81.87 (0.25) | 79.48 (0.32) |
| JTT | For $y_{\mathrm{mt}}=0$ and $y_{\mathrm{sp}}=0$ | 79.66 (0.33) | 78.52 (0.29) | 78.66 (0.34) | 79.67 (0.48) | 82.96 (0.56) |
| | For $y_{\mathrm{mt}}=0$ and $y_{\mathrm{sp}}=1$ | 85.97 (0.31) | 86.65 (0.29) | 86.70 (0.25) | 84.78 (0.57) | 78.26 (1.14) |
| | For $y_{\mathrm{mt}}=1$ and $y_{\mathrm{sp}}=0$ | 85.42 (0.36) | 86.13 (0.34) | 85.97 (0.30) | 84.26 (0.67) | 78.29 (1.11) |
| | For $y_{\mathrm{mt}}=1$ and $y_{\mathrm{sp}}=1$ | 79.71 (0.36) | 78.64 (0.32) | 78.72 (0.32) | 79.92 (0.46) | 83.64 (0.57) |
| | Worst-group | 77.30 (0.25) | 76.71 (0.25) | 76.66 (0.23) | 75.33 (0.40) | 72.32 (0.90) |
| | Average | 82.70 (0.09) | 82.48 (0.09) | 82.50 (0.09) | 82.17 (0.13) | 80.81 (0.32) |

Table 10: **Results of instance reweighing methods for the Toy dataset for** $\rho \in \{0.5, 0.6, 0.7, 0.8, 0.9\}$. Table shows the average, worst-group, and per-group accuracy on a test set without spurious correlation, as a function of the spurious correlation in the training data. Each accuracy is obtained by averaging over 100 runs. Standard error is reported between brackets.

| Method | Accuracy | $p_{\mathrm{train}}(y_m = y\|y_c = y)$ 0.95 |
|---|---|---|
| JSE | $y_{\mathrm{mt}} = 0, y_{\mathrm{sp}} = 0$ | 91.66 (0.59) |
| | $y_{\mathrm{mt}} = 0, y_{\mathrm{sp}} = 1$ | 78.49 (1.09) |
| | $y_{\mathrm{mt}} = 1, y_{\mathrm{sp}} = 0$ | 89.44 (0.43) |
| | $y_{\mathrm{mt}} = 1, y_{\mathrm{sp}} = 1$ | 93.72 (0.25) |
| | Worst-group | 78.49 (1.09) |
| | Average | 86.52 (0.50 |
| GW-ERM | $y_{\mathrm{mt}} = 0, y_{\mathrm{sp}} = 0$ | 89.84 (0.66) |
| | $y_{\mathrm{mt}} = 0, y_{\mathrm{sp}} = 1$ | 87.38 (0.19) |
| | $y_{\mathrm{mt}} = 1, y_{\mathrm{sp}} = 0$ | 90.56 (0.33) |
| | $y_{\mathrm{mt}} = 1, y_{\mathrm{sp}} = 1$ | 90.93 (0.09) |
| | Worst-group | 89.08 (0.26) |
| | Average | 89.08 (0.26) |
| GDRO | $y_{\mathrm{mt}} = 0, y_{\mathrm{sp}} = 0$ | 92.08 (0.59) |
| | $y_{\mathrm{mt}} = 0, y_{\mathrm{sp}} = 1$ | 84.87 (0.84) |
| | $y_{\mathrm{mt}} = 1, y_{\mathrm{sp}} = 0$ | 88.71 (0.54) |
| | $y_{\mathrm{mt}} = 1, y_{\mathrm{sp}} = 1$ | 91.71 (0.37) |
| | Worst-group | 84.66 (0.79) |
| | Average | 88.86 (0.25) |
| JTT | $y_{\mathrm{mt}} = 0, y_{\mathrm{sp}} = 0$ | 97.24 (0.22) |
| | $y_{\mathrm{mt}} = 0, y_{\mathrm{sp}} = 1$ | 54.55 (2.56) |
| | $y_{\mathrm{mt}} = 1, y_{\mathrm{sp}} = 0$ | 78.58 (0.79) |
| | $y_{\mathrm{mt}} = 1, y_{\mathrm{sp}} = 1$ | 96.67 (0.18) |
| | Worst-group | 54.55 (2.56) |
| | Average | 78.50 (1.01) |

Table 11: **Results of instance reweighing methods and JSE for the Waterbirds dataset when** $p_{\mathrm{train}}(y_{\mathrm{mt}} = y\|y_{\mathrm{sp}} = y) = 0.95$: Table shows the average, worst-group, and per-group accuracy on a test set where $p_{\mathrm{OOD}}(y_{\mathrm{mt}} = y\|y_{\mathrm{sp}} = y) = 0.5$, with $y \in \{0, 1\}$. Each accuracy is obtained by averaging over 20 runs. Standard error is reported between brackets.

## B ADDITIONAL RESULTS FOR TOY DATASET

### B.1 REMOVING THE ORTHOGONALITY ASSUMPTION

In this section, we provide an analysis on how JSE performs when the $\mathcal{Z}_{\text{sp}}$ and $\mathcal{Z}_{\text{mt}}$ are not orthogonal subspaces. We create an example of the Toy dataset where the orthogonality assumption does not hold, by changing the angle of $\boldsymbol{w}_{\text{sp}}$ and $\boldsymbol{w}_{\text{mt}}$ to $75°$. Let $a = \cos(\frac{15\pi}{180}), b = \sin(\frac{15\pi}{180})$, and $\boldsymbol{w}_{\text{sp}} = (\gamma, 0, 0, \ldots, 0)^T$ and $\boldsymbol{w}_{\text{mt}} = \left(\frac{\gamma}{1+\frac{a}{b}}, \frac{\gamma}{1+\frac{b}{a}}, 0 \ldots, 0\right)^T$. The main-task labels are now determined by a linear combination of the spurious and main-task directions. In Figure 6, we illustrate the subspaces found by JSE for this scenario. JSE finds a spurious and main-task vector that are slightly different from the basis vectors, yet orthogonal. This illustrates that when the $\mathcal{Z}_{\text{sp}}$ and $\mathcal{Z}_{\text{mt}}$ are not orthogonal subspaces, JSE finds two orthogonal subspaces that best fit given the data.

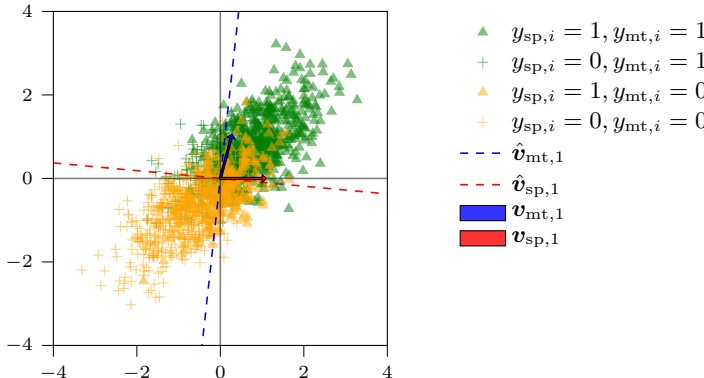

Figure 6: **Illustration of vectors found by JSE with non-orthogonal spurious and main-task subspaces**: The blue and red arrow indicate the basis of the spurious and main-task subspaces, which have an angle of $75°$. Data is based on a single simulation from the $d(=20)$-dimensional Toy dataset of Section 4.1 with $\rho = 0.8$ and sample size $n = 2,000$.

Figure 7 shows the performance for the toy dataset when the subspaces are not orthogonal. Compared to the case where the subspaces are orthogonal, the performance of JSE is slightly worse. It removes a small part of the main-task direction, and leaves a small part of the spurious direction. However, it still outperforms other concept-removal methods such as INLP and RLACE. These methods perform relatively worse, because they also remove part of the main-task direction - even when there is no correlation between the spurious and main-task direction. When the spurious direction is removed, the part of the main-task direction that is non-orthogonal to it is also removed.

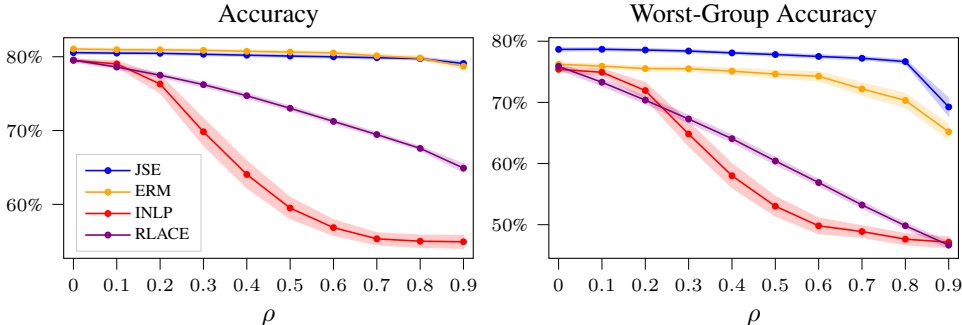

Figure 7: **OOD generalization for Toy dataset, when the angle between the spurious and main-task vector is** $75°$: We plot the (worst-group) accuracy on a test set without spurious correlation, as a function of the spurious correlation in the training data. Each accuracy is obtained by averaging over 100 runs. The shaded area reflects the 95% confidence interval.

## B.2 Finite-sample estimation noise for JSE and INLP

In this section, we briefly illustrate how the performance of JSE and INLP is affected by the interaction of (i) the size of the training set, and (ii) the correlation between the spurious and main-task features. Figure 8 shows the result of applying JSE and INLP to the Toy dataset for different sizes of the dataset. For JSE, we observe that with a limited sample size and a high spurious correlation, it is harder to separate the spurious and main-task features. We attribute this to finite-sample noise: our method finds two orthogonal vectors that fit well in the training data, but they are less likely to align with the data-generating process. Interestingly, INLP becomes worse as the sample size increases. With a greater sample size, it is more likely that INLP assigns the main-task feature as belonging to the spurious subspace, since its predictive ability of the spurious concept label is more likely to be detected.

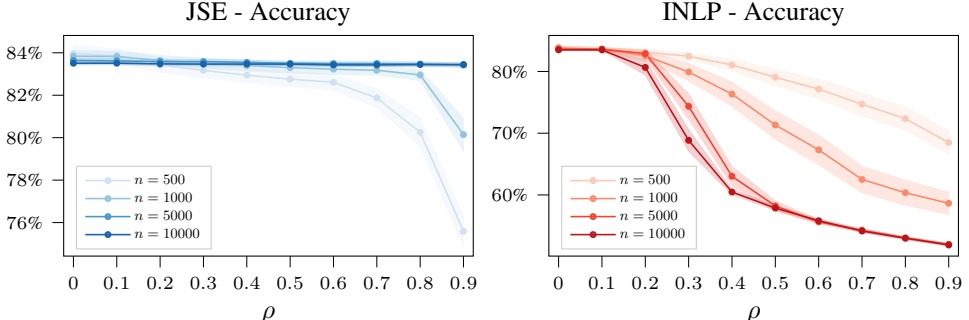

Figure 8: **Effect of training set size and spurious correlation strength for Toy dataset**: We plot the accuracy on a test set without spurious correlation, as a function of the spurious correlation in the training data. Each accuracy is obtained by averaging over 100 runs. The shaded area reflects the 95% confidence interval.

## C  The JSE algorithm

In this section, we first provide a more detailed version of the JSE algorithm, initially described in Section 3.2. We briefly investigate two modifications of the algorithm.

### C.1  Detailed description of the JSE algorithm

In Equation 2 we note that we optimize over the coefficients for two logistic regressions, while enforcing an orthogonality constraint. Here, we briefly explain how this constraint is enforced. We solve the following unconstrained problem using stochastic gradient descent (SGD):

$$\hat{\boldsymbol{w}}_{\mathrm{sp}}, \hat{\boldsymbol{w}}_{\mathrm{mt}}, \hat{b}_{\mathrm{sp}}, \hat{b}_{\mathrm{mt}} = \underset{\boldsymbol{w}_{\mathrm{sp}}, \boldsymbol{w}_{\mathrm{mt}}, b_{\mathrm{sp}}, b_{\mathrm{mt}}}{\arg\min} \sum_{i=1}^{n} \mathcal{L}_{\mathrm{BCE}}(\hat{y}_{\mathrm{sp},i}, y_{\mathrm{sp},i}) + \mathcal{L}_{\mathrm{BCE}}(\hat{y}_{\mathrm{mt},i}, y_{\mathrm{mt},i}),$$

where we define the predictions as follows:

$$\hat{y}_{\mathrm{sp},i} = \mathrm{Logit}^{-1}\left(\boldsymbol{z}_i^T \boldsymbol{w}_{\mathrm{sp}} + b_{\mathrm{sp}}\right),$$
$$\hat{y}_{\mathrm{mt},i} = \mathrm{Logit}^{-1}\left(\boldsymbol{z}_i^T (\boldsymbol{I} - \boldsymbol{P}_{\boldsymbol{w}_{\mathrm{sp}}})\boldsymbol{w}_{\mathrm{mt}} + b_{\mathrm{mt}}\right),$$
$$\boldsymbol{P}_{\boldsymbol{w}_{\mathrm{sp}}} = \boldsymbol{w}_{\mathrm{sp}}\left(\boldsymbol{w}_{\mathrm{sp}}^T \boldsymbol{w}_{\mathrm{sp}}\right)^{-1}\boldsymbol{w}_{\mathrm{sp}}^T.$$

By definition, $(\boldsymbol{I} - \boldsymbol{P}_{\boldsymbol{w}_{\mathrm{sp}}})\boldsymbol{w}_{\mathrm{mt}}$ is orthogonal to $\boldsymbol{w}_{\mathrm{sp}}$. Therefore, the predictions for each $y_{\mathrm{mt},i}$ are based upon a set of coefficients that is orthogonal to $\boldsymbol{w}_{\mathrm{sp}}$. By weighing both losses equally, there is no reason for this optimization problem to favor the loss for one set of labels over the other.

In Algorithm 2 we provide the exact same procedure as in Algorithm 1, but in greater detail.

---

**Algorithm 2** JSE algorithm to estimate orthonormal bases for $\mathcal{Z}_{\text{sp}}$ and $\mathcal{Z}_{\text{mt}}$. The calculation of the test statistics is discussed in Section D.3

---

**Require:** a sample $\{y_{\text{mt},i}, y_{\text{sp},i}, \boldsymbol{z}_i\}_{i=1}^n$ consisting of two binary labels and a vector $\boldsymbol{z}_i \in \mathbb{R}^d$.

Initialize a $(n \times d)$-dimensional embedding matrix $\boldsymbol{Z} = (\boldsymbol{z}_1 \, \boldsymbol{z}_2 \, \cdots \, \boldsymbol{z}_n)^T$.

Initialize $\boldsymbol{Z}_{\text{sp}}^{\perp} \leftarrow \boldsymbol{Z}$.

Choose a significance level $\alpha$ for the test statistics, resulting in critical value $t_{1-\alpha}$

**for** $i = 1, ..., d$ **do**

   $\boldsymbol{Z}_{\text{remain}} \leftarrow \boldsymbol{Z}_{\text{sp}}^{\perp}$

   **for** $j = 1, ..., d$ **do**

      Estimate $\hat{\boldsymbol{w}}_{\text{sp}}$ and $\hat{\boldsymbol{w}}_{\text{mt}}$ with Equation 2, using embeddings $\boldsymbol{Z}_{\text{remain}}$.

      Define normalizations $\hat{\boldsymbol{v}}_{\text{sp},i} \leftarrow \hat{\boldsymbol{w}}_{\text{sp}}/||\hat{\boldsymbol{w}}_{\text{sp}}||$ and $\hat{\boldsymbol{v}}_{\text{mt},j} \leftarrow \hat{\boldsymbol{w}}_{\text{mt}}/||\hat{\boldsymbol{w}}_{\text{mt}}||$.

      Estimate $\hat{\gamma}_{\text{sp}}, \hat{b}_{\text{sp}} = \underset{\gamma_{\text{sp}}, b_{\text{sp}}}{\arg\min} \sum_{h=1}^n \mathcal{L}_{\text{BCE}}(\hat{y}_{\text{sp},h}^{(\hat{\boldsymbol{v}}_{\text{mt},j})}, y_{\text{sp},h})$,

            where $\hat{y}_{\text{sp},h}^{(\hat{\boldsymbol{v}}_{\text{mt},j})} = \text{Logit}^{-1}\left(\gamma_{\text{sp}} \boldsymbol{z}_{h,\text{remain}}^T \hat{\boldsymbol{v}}_{\text{mt},j} + b_{\text{sp}}\right)$

      Calculate $t_{\text{mt,rnd}}, t_{\text{mt,sp}}^{(\hat{\boldsymbol{v}}_{\text{mt},j})}$ (see Section D.3)

      **if** $(t_{\text{mt,rnd}} < -t_{1-\alpha})$ and $(t_{\text{mt,sp}}^{(\hat{\boldsymbol{v}}_{\text{mt},j})} > t_{1-\alpha})$ **then**

         Projection $\boldsymbol{Z}_{\text{remain}} \leftarrow \boldsymbol{Z}_{\text{sp}}^{\perp}(\boldsymbol{I} - \hat{\boldsymbol{V}}_{\text{mt}} \hat{\boldsymbol{V}}_{\text{mt}}^T)$, where $\hat{\boldsymbol{V}}_{\text{mt}} = (\hat{\boldsymbol{v}}_{\text{mt},1} \, \hat{\boldsymbol{v}}_{\text{mt},2} \, \cdots \, \hat{\boldsymbol{v}}_{\text{mt},j})$

      **else**

         **break**

      **end if**

   **end for**

   Estimate $\hat{\gamma}_{\text{mt}}, \hat{b}_{\text{mt}} = \sum_{h=1}^n \underset{\gamma_{\text{mt}}, b_{\text{mt}}}{\arg\min} \mathcal{L}_{\text{BCE}}(\hat{y}_{\text{mt},h}^{(\hat{\boldsymbol{v}}_{\text{sp},i})}, y_{\text{mt},h})$,

         where $\hat{y}_{\text{mt},h}^{(\hat{\boldsymbol{v}}_{\text{sp},i})} = \text{Logit}^{-1}\left(\gamma_{\text{mt}} \boldsymbol{z}_{h,\text{sp}}^{\perp\,T} \hat{\boldsymbol{v}}_{\text{sp},i} + b_{\text{sp}}\right)$

   Calculate $t_{\text{sp,rnd}}, t_{\text{mt,sp}}^{(\hat{\boldsymbol{v}}_{\text{sp},i})}$ (see Section D.3)

   **if** $(t_{\text{sp,rnd}} < -t_{1-\alpha})$ and $(t_{\text{mt,sp}}^{(\hat{\boldsymbol{v}}_{\text{sp},i})} < -t_{1-\alpha})$ **then**

      Projection $\boldsymbol{Z}_{\text{sp}}^{\perp} \leftarrow \boldsymbol{Z}(\boldsymbol{I} - \hat{\boldsymbol{V}}_{\text{sp}} \hat{\boldsymbol{V}}_{\text{sp}}^T)$, where $\hat{\boldsymbol{V}}_{\text{sp}} = (\hat{\boldsymbol{v}}_{\text{sp},1} \, \hat{\boldsymbol{v}}_{\text{sp},2} \, \cdots \, \hat{\boldsymbol{v}}_{\text{sp},i})$.

   **else**

      **break**

   **end if**

**end for**

**return** $\hat{\boldsymbol{v}}_{\text{sp},1}, \, \hat{\boldsymbol{v}}_{\text{sp},2}, \, \ldots, \, \hat{\boldsymbol{v}}_{\text{sp},i}$

---

### C.2 MODIFICATIONS OF THE JSE ALGORITHM

In this section, we briefly compare the formulation of the JSE algorithm to two alternatives.

- **Swapping the loops:** one might consider interchanging the inner loop and the outer loop of the JSE algorithm. Now, at each step, first the spurious vectors $\hat{\boldsymbol{v}}_{\text{sp},1}, \hat{\boldsymbol{v}}_{\text{sp},2}, \ldots, \hat{\boldsymbol{v}}_{\text{sp},d_{\text{sp}}}$ are projected out before the main-task vector (at that step) is estimated.

- **Projecting onto main-task subspace**: instead of projecting $\boldsymbol{z}$ onto the orthogonal complement of $\mathcal{Z}_{\text{sp}}$, one could be interested in projecting onto $\mathcal{Z}_{\text{mt}}$. In this case, rather than the transformation $(\boldsymbol{I} - \boldsymbol{V}_{\text{sp}} \boldsymbol{V}_{\text{sp}}^T)\boldsymbol{z}$, one uses the transformed embeddings $(\boldsymbol{V}_{\text{mt}} \boldsymbol{V}_{\text{mt}}^T)\boldsymbol{z}$.

In Table 12 we compare these two alternative formulations of the algorithm for the Waterbirds dataset. The performance for these two versions is similar to the JSE algorithm as outlined in Section

3.2, although we observe a small drop in worst-group accuracy for the version with switched for-loops. We suggest that in practice, which version of the algorithm should be used could depend on (1) whether the user wants to remove a certain spurious concept, or isolate certain main-task features, and (2) the dimensionality of both subspaces. For example, if the dimension of the main-task subspace is much lower than that of the spurious concept subspace, it might be worthwhile to project onto $z_{\mathrm{mt}}$ to improve the bias-variance trade-off, rather than removing $z_{\mathrm{sp}}$.

| | | $p_{\mathrm{train}}(y_{\mathrm{mt}} = y\|y_{\mathrm{sp}} = y)$ | | | | |
|---|---|---|---|---|---|---|
| Method | Accuracy | 0.5 | 0.6 | 0.7 | 0.8 | 0.9 |
| JSE with projecting onto $z_{\mathrm{mt}}$ | $y_{\mathrm{mt}} = 0, y_{\mathrm{sp}} = 0$ | 93.40 (0.15) | 91.55 (0.80) | 91.44 (0.37) | 91.60 (0.31) | 89.24 (0.87) |
| | $y_{\mathrm{mt}} = 0, y_{\mathrm{sp}} = 1$ | 89.81 (0.20) | 87.69 (0.98) | 87.79 (0.46) | 89.62 (0.40) | 86.72 (1.00) |
| | $y_{\mathrm{mt}} = 1, y_{\mathrm{sp}} = 0$ | 90.58 (0.11) | 91.60 (0.26) | 91.89 (0.15) | 91.64 (0.14) | 92.00 (0.22) |
| | $y_{\mathrm{mt}} = 1, y_{\mathrm{sp}} = 1$ | 90.99 (0.16) | 91.63 (0.25) | 91.84 (0.16) | 91.47 (0.13) | 90.94 (0.33) |
| | Worst-group | 89.63 (0.17) | 87.67 (0.97) | 87.79 (0.46) | 89.53 (0.38) | 86.50 (0.96) |
| | Average | 91.42 (0.10) | 90.06 (0.64) | 90.11 (0.29) | 90.82 (0.23) | 88.75 (0.64) |
| JSE with Swapped Loops | $y_{\mathrm{mt}} = 0, y_{\mathrm{sp}} = 0$ | 92.77 (0.16) | 90.88 (0.65) | 90.26 (0.34) | 91.58 (0.48) | 93.13 (0.23) |
| | $y_{\mathrm{mt}} = 0, y_{\mathrm{sp}} = 1$ | 89.50 (0.23) | 88.31 (0.68) | 88.08 (0.56) | 87.84 (0.38) | 83.11 (0.95) |
| | $y_{\mathrm{mt}} = 1, y_{\mathrm{sp}} = 0$ | 90.76 (0.13) | 91.63 (0.24) | 92.02 (0.10) | 91.12 (0.24) | 90.13 (0.21) |
| | $y_{\mathrm{mt}} = 1, y_{\mathrm{sp}} = 1$ | 91.71 (0.14) | 91.79 (0.21) | 91.88 (0.24) | 92.22 (0.26) | 93.00 (0.33) |
| | Worst-group | 89.42 (0.22) | 88.26 (0.68) | 88.01 (0.55) | 87.47 (0.35) | 83.10 (0.94) |
| | Average | 91.16 (0.12) | 90.06 (0.48) | 89.79 (0.28) | 90.15 (0.16) | 88.89 (0.32) |

Table 12: **Results for the Waterbirds dataset for different versions of the JSE algorithm**: Table shows the average, worst-group, and per-group accuracy on a test set where $p_{\mathrm{OOD}}(y_{\mathrm{mt}} = y\|y_{\mathrm{sp}} = y) = 0.5$, with $y \in \{0, 1\}$. Each accuracy is obtained by averaging over 20 runs. Standard error is reported between brackets.

## D    DETAILS ON THE TESTING PROCEDURE OF JSE

In this section, we provide a detailed explanation of the testing procedure used in the JSE algorithm for breaking the for-loops. Section D.1 will introduce notation, and a general set-up that is applicable for each test. In Section D.2 we provide the derivations that are needed for our subsequent test statistics. In Section D.3 we define the test statistics that are used throughout the JSE method. In Section D.4 we study the effect of our choice to equally weight across the four groups in the data when measuring the test statistics, and in Section D.5 we investigate how one can adjust the test in cases where one set of labels is much harder to predict than the other.

### D.1    NOTATION AND SET-UP

Recall from Section 3.3 that we are interested in testing the difference between two binary cross-entropies (BCE). For example, we can define the difference in BCE between a logistic regression that is trained on our embeddings $z$ to predict the spurious concept labels $y_{\mathrm{sp}}$ and one that just uses an intercept (referred to as 'random' classifier).

For a particular sample, one could simply estimate this difference by weighting the individual observations equally. However, for our tests, we noticed that there was a considerable improvement if the difference was weighted equally within the subgroups specified by the pair of labels $(y_{\mathrm{mt}}, y_{\mathrm{sp}})$. This is examined in Section D.4. Below, we formulate how one, in general, can test for such a weighted difference between BCE's of two classifiers.

Based on the main task and spurious labels $y_{\mathrm{mt}}, y_{\mathrm{sp}}$, we define four groups: (1) $y_{\mathrm{mt}} = 0, y_{\mathrm{sp}} = 0$, (2) $y_{\mathrm{mt}} = 0, y_{\mathrm{sp}} = 1$, (3) $y_{\mathrm{mt}} = 1, y_{\mathrm{sp}} = 0$, and (4) $y_{\mathrm{mt}} = 1, y_{\mathrm{sp}} = 1$. The spurious and main-task labels are used to define a new random variable $G \in \{1, 2, 3, 4\}$, indicating group membership. The probability of a group $g$ is noted as $\pi_g$.

We consider the difference between the binary cross-entropy (BCE) as a random variable $d$. We assume that this difference is a mixture of four random variables: $d_1, d_2, d_3, d_4$. The random variable $d_g$ corresponds to the difference in BCE for group $g$. We assume that the expectation and variance of each of these four random variables is different, e.g. $\mathbb{E}[d_g] \neq \mathbb{E}[d_h]$, $\mathrm{Var}[d_g] \neq \mathrm{Var}[d_h]$, for $g \neq h$. However, we do assume that they are independent: $\mathrm{Cov}(d_g, d_h) = 0$. I denote the $\mathbb{E}[d_g] = \mu_g$ and $\mathrm{Var}[d_g] = \sigma_g^2$.

We are interested in the weighted average of $d_1, d_2, d_3, d_4$, where each group receives an equal weight. Concretely, we can define a new random variable

$$d_w = \frac{1}{4} \sum_{g=1}^{4} d_g.$$

The subscript of $w$ will be used to refer to the equally weighted sum of difference $d$. We are interested in the following hypotheses

$$H_0 : \mathbb{E}[d_w] = 0, \quad H_1 : \mathbb{E}[d_w] < 0.$$

We draw a sample of $n$ independent and identical (IID) observations from $d$ and $G$.

Because we observe $G$, we know for each observation which random variable we are observing - e.g. if $G = 1$, we observe $d_1$. This means that after drawing the $n$ observations, we observe the $n_g$ observations for group $g$. The observations of $d$ from group $g$ are denoted as $d_{g,i}$ for $i = 1, 2, ..., n_g$. However, we do not know the value of each $\pi_g$. We do assume each $\pi_g$ is strictly positive.

### D.2 DERIVATION OF TEST STATISTIC

The expectation of $d_w$ is

$$\mu_w = \mathbb{E}[d_w] = \frac{1}{4}\mathbb{E}\left[\sum_{g=1}^{4} d_g\right] = \frac{1}{4} \sum_{g=1}^{4} \mu_g.$$

Let $\bar{d}_w$ denote an estimator of $d_w$

$$\bar{d}_w = \frac{1}{4} \sum_{g=1}^{4} \bar{d}_g, \text{ with } \bar{d}_g = \frac{1}{n_g} \sum_{i=1}^{n_g} d_{i,g},$$

where $n_g$ is a random variable. We can show this estimator is unbiased via the law of total expectation.

$$
\begin{aligned}
\mathbb{E}[\bar{d}_g] &= \mathbb{E}\left[\frac{1}{n_g} \sum_{i=1}^{n_g} d_{i,g}\right] \\
&= \mathbb{E}\left[\mathbb{E}\left[\frac{1}{n_g} \sum_{i=1}^{n_g} d_{i,g}|n_g\right]\right] \\
&= \mathbb{E}\left[\frac{1}{n_g} \sum_{i=1}^{n_g} \mathbb{E}\left[d_{i,g}|n_g\right]\right] \\
&= \mathbb{E}\left[\frac{1}{n_g} n_g \mu_g\right] \quad &\text{(Since they are IID)} \\
&= \mu_g, \\
\mathbb{E}[\bar{d}_w] &= \frac{1}{4} \sum_{g=1}^{4} \mu_g.
\end{aligned}
$$

We now turn to the variance of $\bar{d}_w$

$$
\begin{aligned}
\text{Var}(\bar{d}_w) &= \text{Var}\left(\frac{1}{4} \sum_{g=1}^{4} \bar{d}_g\right) \\
&= \frac{1}{16}\text{Var}\left(\sum_{g=1}^{4} \bar{d}_g\right) \\
&= \frac{1}{16} \sum_{g=1}^{4} \sum_{h=1}^{4} \text{Cov}(\bar{d}_g, \bar{d}_h).
\end{aligned}
$$

First, it is shown first show that $\text{Cov}(\bar{d}_g, \bar{d}_h) = 0$ for $g \neq h$ via the law of total covariance:

$$\text{Cov}(\bar{d}_g, \bar{d}_h) = \mathbb{E}[\text{Cov}(\bar{d}_g, \bar{d}_h|n_g, n_h)] + \text{Cov}(\mathbb{E}[\bar{d}_g|n_g, n_h], \mathbb{E}[\bar{d}_h|n_g, n_h])$$

Since the sample means in expectation are the constants $\mu_g$, $\mu_h$, their covariance is 0:

$$\text{Cov}(\mathbb{E}[\bar{d}_g|n_g, n_h], \mathbb{E}[\bar{d}_h|n_g, n_h]) = \text{Cov}(\mu_g, \mu_h) = 0.$$

Next, we define

$$\mathbb{E}[\text{Cov}(\bar{d}_g, \bar{d}_h|n_g, n_h)] = \mathbb{E}\left[\frac{1}{n_g}\frac{1}{n_h}\text{Cov}\left(\sum_{i=1}^{n_g} d_{i,g}, \sum_{i=1}^{n_h} d_{i,h}|n_g, n_h\right)\right]$$

$$= \mathbb{E}\left[\frac{1}{n_g}\frac{1}{n_h}0\right] = 0,$$

This last step can be made because we assume $d_{g,i}$ is independent of $d_{h,j}$ for $g \neq h$, $i = 1, ..., n_g$ and $j = 1, ..., n_h$. The variance of $\bar{d}_w$ becomes

$$\text{Var}\left(\bar{d}_w\right) = \frac{1}{16}\sum_{g=1}^{4}\text{Var}\left(\bar{d}_g\right).$$

We can define $\text{Var}\left(\bar{d}_g\right)$ via the law of total variance

$$\text{Var}\left(\bar{d}_g\right) = \text{Var}\left(\frac{1}{n_g}\sum_{i=1}^{n_g} d_{i,g}\right)$$

$$= \mathbb{E}\left[\text{Var}\left(\frac{1}{n_g}\sum_{i=1}^{n_g} d_{i,g}|n_g\right)\right] + \text{Var}\left(\mathbb{E}\left[\frac{1}{n_g}\sum_{i=1}^{n_g} d_{i,g}|n_g\right]\right)$$

$$= \mathbb{E}\left[\text{Var}\left(\frac{1}{n_g}\sum_{i=1}^{n_g} d_{i,g}|n_g\right)\right] + \text{Var}\left(\mu_g\right)$$

$$= \mathbb{E}\left[\text{Var}\left(\frac{1}{n_g}\sum_{i=1}^{n_g} d_{i,g}|n_g\right)\right] \qquad \text{(Since variance of constant is 0)}$$

$$= \mathbb{E}\left[\frac{1}{n_g^2}\sum_{i=1}^{n_g}\text{Var}\left(d_{i,g}\right)\right]$$

$$= \mathbb{E}[\frac{1}{n_g^2}n_g\sigma_g^2] \qquad \text{(Since they are IID)}$$

$$= \mathbb{E}[\frac{1}{n_g}]\sigma_g^2.$$

For $n \to \infty$, $n_g$ approximately follows a binomial distribution with $\mathbb{E}[n_g] = n\pi_g$ and variance $n\pi_g(1 - \pi_g)$. We define a second order Taylor expansion to approximate $\frac{1}{n_g}$ around $\mathbb{E}[n_g] = n\pi_g$

$$\mathbb{E}[\frac{1}{n_g}] \approx \frac{1}{n\pi_g} + \frac{(1 - \pi_g)}{n^2\pi_g^2}.$$

This means:

$$\text{Var}\left(\bar{d}_g\right) \approx \left(\frac{1}{n\pi_g} + \frac{(1 - \pi_g)}{n^2\pi_g^2}\right)\sigma_g^2$$

$$= \frac{1}{n\pi_g}\sigma_g^2 + \mathcal{O}(n^{-2}).$$

Using this, we approximate the variance of the weighted sum $d_w$ via

$$\text{Var}(\bar{d}_w) \approx \frac{1}{16}\sum_{g=1}^{4}\frac{1}{n\pi_g}\sigma_g^2.$$

Using the expectation and variance of $\bar{d}_w$, we now proceed to its asymptotic distribution. Assuming that (i) the sample means $\bar{d}_g$ are based on IID random variables and (ii) $\sigma_g^2$ is bounded, we can use the central limit theorem (CLT) for each of the four sample means

$$\sqrt{n}\left(\bar{d}_g - \mu_g\right) \xrightarrow{d} \mathcal{N}(0, \frac{\sigma_g^2}{\pi_g}).$$

This holds because $lim_{n\to\infty} n_g/n = \pi_g > 0$ for all $g$; see for instance Rényi (1957). Given that the CLT holds for each sample mean, joint convergence follows by independence of the sample means. Hence, the distribution of the linear combination directly follows

$$\sqrt{n}(\bar{d}_w - \mathbb{E}[d_w]) = \frac{1}{4} \sum_{g=1}^{4} \sqrt{n}(\bar{d}_g - \mu_g) \xrightarrow{d} \mathcal{N}\left(0, \frac{1}{16} \sum_{g=1}^{4} \frac{\sigma_g^2}{\pi_g}\right).$$

Hence, the variance of $\bar{d}_w$

$$\mathrm{Var}(\bar{d}_w) \approx \frac{1}{16} \sum_{g=1}^{4} \frac{\sigma_g^2}{n\pi_g}$$

can be consistently estimated via

$$\widehat{\mathrm{Var}}(\bar{d}_w) = \frac{1}{16} \sum_{g=1}^{4} \frac{s_g^2}{n\hat{\pi}_g} = \frac{1}{16} \sum_{g=1}^{4} \frac{s_g^2}{n_g},$$

with $\hat{\pi}_g = n_g/n$, and $s_g^2 = \frac{1}{n_g-1} \sum_{i=1}^{n_g} (d_{g,i} - \bar{d}_g)^2$. Using this, we can define a test statistic $t_w$, which for $n \to \infty$

$$t_w = \frac{\bar{d}_w - \mathbb{E}[d_w]}{\widehat{\mathrm{Var}}(\bar{d}_w)} \xrightarrow{d} \mathcal{N}(0, 1).$$

### D.3 TEST STATISTICS FOR JSE

In the previous section, we defined a test statistic for the equally weighted weighted average of $d$. Here, we will use this derivation for the test statistics in the inner and outer loop of JSE.

We start with the first criterion, namely that the vector $\boldsymbol{v}_{\mathrm{sp}}$ ($\boldsymbol{v}_{\mathrm{mt}}$) is informative about the spurious label (main-task label). This criterion is operationalised as follows: the $\boldsymbol{v}_{\mathrm{sp}}$ should contain more information about $y_{\mathrm{sp}}$ than a majority-rule 'random classifier'. The coefficients for a logistic regression can be written as a combination of a unit vector and a scalar: $\boldsymbol{w} = \boldsymbol{v}\gamma$. Consider a logistic regression model $\hat{y}_{\mathrm{sp}}^{(\boldsymbol{v}_{\mathrm{sp}})} = \mathrm{Logit}^{-1}\left(\gamma_{\mathrm{sp}}\boldsymbol{z}^T\boldsymbol{v}_{\mathrm{sp}} + b_{\mathrm{sp}}\right)$. This is a predictor for the label $y_{\mathrm{sp}}$ based on the embeddings projected onto $\boldsymbol{v}_{\mathrm{sp}}$. Let $\hat{y}_{\mathrm{sp}}^{(\mathrm{rnd})}$ denote a random classifier. We can define the difference between these two classifiers as

$$d_{\mathrm{sp,rnd}}^{(\boldsymbol{v}_{\mathrm{sp}})} = \mathcal{L}_{\mathrm{BCE}}(\hat{y}_{\mathrm{sp}}^{(\boldsymbol{v}_{\mathrm{sp}})}, y_{\mathrm{sp}}) - \mathcal{L}_{\mathrm{BCE}}(\hat{y}_{\mathrm{sp}}^{(\mathrm{rnd})}, y_{\mathrm{sp}}).$$

The first criterion translates into the following hypothesis test

$$H_0 : \mathbb{E}[d_{w,\mathrm{sp,rnd}}^{(\boldsymbol{v}_{\mathrm{sp}})}] = 0 \quad \text{versus} \quad H_1 : \mathbb{E}[d_{w,\mathrm{sp,rnd}}^{(\boldsymbol{v}_{\mathrm{sp}})}] < 0,$$

which means that under the null hypothesis, there is no difference in the BCE of these two classifiers. Under the alternative hypothesis, the BCE of a random classifier is higher.

We use the following test statistic, where under the null hypothesis

$$t_{\mathrm{sp,rnd}} = \frac{\bar{d}_{w,\mathrm{sp,rnd}}^{(\boldsymbol{v}_{\mathrm{sp}})}}{\mathrm{Var}(\bar{d}_{w,\mathrm{sp,rnd}}^{(\boldsymbol{v}_{\mathrm{sp}})})} \xrightarrow{d} \mathcal{N}(0, 1).$$

The previous test can also be defined for a main-task vector $\boldsymbol{v}_{\mathrm{mt}}$ and main-task labels $y_{\mathrm{mt}}$. Consider a logistic regression model $\hat{y}_{\mathrm{mt}}^{(\boldsymbol{v}_{\mathrm{mt}})} = \mathrm{Logit}^{-1}\left(\gamma_{\mathrm{mt}}\boldsymbol{z}^T\boldsymbol{v}_{\mathrm{mt}} + b_{\mathrm{mt}}\right)$. The difference between the BCE of this classifier and a random classifier $\hat{y}_{\mathrm{mt}}^{(\mathrm{rnd})}$ is:

$$d_{\mathrm{mt,rnd}}^{(\boldsymbol{v}_{\mathrm{mt}})} = \mathcal{L}_{\mathrm{BCE}}(\hat{y}_{\mathrm{mt}}^{(\boldsymbol{v}_{\mathrm{mt}})}, y_{\mathrm{mt}}) - \mathcal{L}_{\mathrm{BCE}}(\hat{y}_{\mathrm{mt}}^{(\mathrm{rnd})}, y_{\mathrm{mt}}),$$

which we can use for the following hypotheses

$$H_0 : \mathbb{E}[d_{w,\text{mt},\text{rnd}}^{(\boldsymbol{v}_{\text{mt}})}] = 0 \quad \text{versus} \quad H_1 : \mathbb{E}[d_{w,\text{mt},\text{rnd}}^{(\boldsymbol{v}_{\text{mt}})}] < 0.$$

For these hypotheses, we define a test statistic similar to the previous one, only now for the main-task vector and labels

$$t_{\text{mt},\text{rnd}} = \frac{\bar{d}_{w,\text{mt},\text{rnd}}^{(\boldsymbol{v}_{\text{mt}})}}{\text{Var}(\bar{d}_{w,\text{mt},\text{rnd}}^{(\boldsymbol{v}_{\text{mt}})})} \xrightarrow{d} \mathcal{N}(0,1).$$

We now turn to the second criterion, which is that the $\boldsymbol{v}_{\text{sp}}$ is more predictive of the spurious concept than the main-task concept (and vice-versa for $\boldsymbol{v}_{\text{mt}}$). This criterion is operationalised as follows: The BCE of a spurious vector $\boldsymbol{v}_{\text{sp}}$ should be lower for the spurious concept than the main-task, and the vice-versa. We compare the BCE's of $\hat{y}_{\text{sp}}^{(\boldsymbol{v}_{\text{sp}})}$ and $\hat{y}_{\text{mt}}^{(\boldsymbol{v}_{\text{sp}})} = \text{Logit}^{-1}\left(\gamma'_{\text{mt}} \boldsymbol{z}^T \boldsymbol{v}_{\text{sp}} + b'_{\text{mt}}\right)$, where the latter is a predictor for the main-task label, based on the embeddings projected onto $\boldsymbol{v}_{\text{sp}}$. The model parameters $\gamma_{\text{mt}}$ and $b_{\text{mt}}$ are to be trained by minimizing the BCE. We define the difference

$$d_{\text{sp},\text{mt}}^{(\boldsymbol{v}_{\text{sp}})} = \mathcal{L}_{\text{BCE}}(\hat{y}_{\text{sp}}^{(\boldsymbol{v}_{\text{sp}})}, y_{\text{sp}}) - \mathcal{L}_{\text{BCE}}(\hat{y}_{\text{mt}}^{(\boldsymbol{v}_{\text{sp}})}, y_{\text{mt}}),$$

and use this difference to test the following hypotheses

$$H_0 : \mathbb{E}[d_{w,\text{sp},\text{mt}}^{(\boldsymbol{v}_{\text{sp}})}] = \Delta \quad \text{versus} \quad H_1 : \mathbb{E}[d_{w,\text{sp},\text{mt}}^{(\boldsymbol{v}_{\text{sp}})}] < \Delta,$$

where the parameter $\Delta$ can be used to adjust for the fact that one label is harder to predict than the other. We give an example of its usefulness in Appendix D.5, and a heuristic for setting the parameter value. Under the null hypothesis, the difference in the BCE for the spurious concept and main-task labels, for a logistic regression based on the embeddings projected onto $\boldsymbol{v}_{\text{sp}}$, is $\Delta$. We can use the following test statistic, where under the null hypothesis

$$t_{\text{sp},\text{mt}}^{(\boldsymbol{v}_{\text{sp}})} = \frac{\bar{d}_{w,\text{sp},\text{mt}}^{(\boldsymbol{v}_{\text{sp}})} - \Delta}{\text{Var}(\bar{d}_{w,\text{sp},\text{mt}}^{(\boldsymbol{v}_{\text{sp}})})} \xrightarrow{d} \mathcal{N}(0,1).$$

We can then conduct the same test, but now for the main-task vector $\boldsymbol{v}_{\text{mt}}$ instead of $\boldsymbol{v}_{\text{sp}}$. Define $\hat{y}_{\text{sp}}^{(\boldsymbol{v}_{\text{mt}})} = \text{Logit}^{-1}\left(\gamma'_{\text{sp}} \boldsymbol{z}^T \boldsymbol{v}_{\text{mt}} + b'_{\text{sp}}\right)$. This test uses the following difference

$$d_{\text{sp},\text{mt}}^{(\boldsymbol{v}_{\text{mt}})} = \mathcal{L}_{\text{BCE}}(\hat{y}_{\text{sp}}^{(\boldsymbol{v}_{\text{mt}})}, y_{\text{sp}}) - \mathcal{L}_{\text{BCE}}(\hat{y}_{\text{mt}}^{(\boldsymbol{v}_{\text{mt}})}, y_{\text{mt}}),$$

and the hypotheses become

$$H_0 : \mathbb{E}[d_{w,\text{sp},\text{mt}}^{(\boldsymbol{v}_{\text{mt}})}] = \Delta \quad \text{versus} \quad H_1 : \mathbb{E}[d_{w,\text{sp},\text{mt}}^{(\boldsymbol{v}_{\text{mt}})}] > \Delta.$$

Under $H_1$ we now test if the difference is greater than $\Delta$ compared to the previous test. We use the following test statistic:

$$t_{\text{sp},\text{mt}}^{(\boldsymbol{v}_{\text{mt}})} = \frac{\bar{d}_{w,\text{sp},\text{mt}}^{(\boldsymbol{v}_{\text{mt}})} - \Delta}{\text{Var}(\bar{d}_{w,\text{sp},\text{mt}}^{(\boldsymbol{v}_{\text{mt}})})} \xrightarrow{d} \mathcal{N}(0,1).$$

For each of the test statistics, given a large enough sample size, we can acquire our critical values from the standard normal distribution for a given significance level $\alpha$.

In general, we used the validation set for the test statistics in order to mitigate the effect of overfitting. Unless otherwise mentioned, we set $\alpha = 0.05$ and $\Delta = 0$.

### D.4 WEIGHTED AVERAGE VS. UNWEIGHTED AVERAGE FOR TEST STATISTICS OF JSE

As stated in the previous section, we use a weighted average for the test statistics of JSE, where the difference in BCE's is weighted equally across the four combinations of $y_{\text{sp}}$ and $y_{\text{mt}}$. We argue that this is helpful in distinguishing whether or not a spurious concept vector $\boldsymbol{v}_{\text{sp}}$ contains a spurious concept or not (vice versa for $\boldsymbol{v}_{\text{mt}}$).

For example, consider a sample where 90% of the water or landbirds coincide with a water or land background ($p_{\text{train}}(y_{\text{mt}} = y|y_{\text{sp}} = y) = 0.9$), and we are interested in determining if a vector $v$ contains information about the spurious or main-task features. If $v$ contains information about the background features, a logistic regression for the label $y_{\text{sp}}$ based on the embeddings projected onto $v$ will have a low BCE for spurious concept labels. However, this logistic regression will likely also have a low BCE for the main-task labels, due to the correlation between the labels. We can address this problem by weighting the BCE equally across the four combinations of $y_{\text{sp}}$ and $y_{\text{mt}}$, where the groups with a small sample (and where $y_{\text{sp}}$ and $y_{\text{mt}}$ do not coincide) will have a greater influence on the overall average

We verify this argument empirically by comparing two versions of JSE: one with the equally weighted average for the test statistics (as presented in this paper), and one which uses a simple average. Figure 9 shows the results of applying these two versions of the JSE algorithm to the Waterbirds dataset. When the spurious correlation is high, the version with a simple average performs worse in terms of both overall and worst-group accuracy.

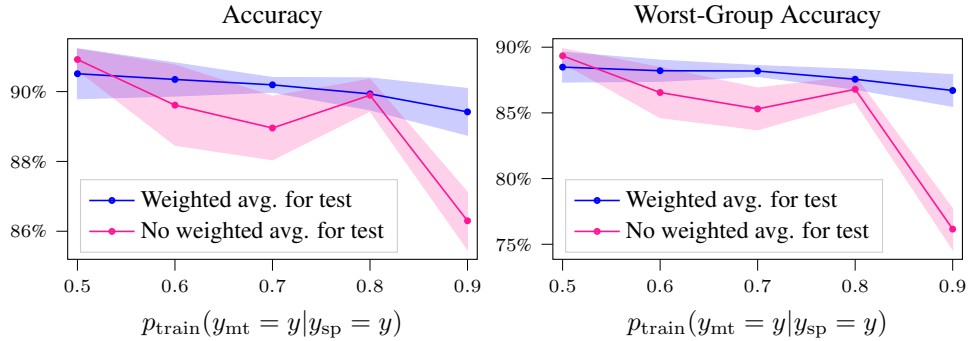

Figure 9: **Effect of using an equally weighted average for the tests of JSE for the Waterbirds dataset**: We plot the (worst-group) accuracy on an OOD test set where $p_{\text{OOD}}(y_{\text{mt}} = y|y_{\text{sp}} = y) = 0.5$, as a function of $p_{\text{train}}(y_{\text{mt}} = y|y_{\text{sp}} = y)$. Each accuracy is obtained by averaging over 20 runs. The shaded area reflects the 95% confidence interval.

### D.5 ADJUSTING FOR DIFFERENT DIFFICULTY IN PREDICTING LABELS

A potential issue with comparing two BCE's is that it might be fundamentally harder to predict one set of labels over the other. Consider the example of distinguishing cows vs. penguins as given in the introduction. If it is much easier to predict the background than the animal type, then we might wrongly attribute main-task vectors to $\mathcal{Z}_{\text{mt}}$. If the spurious concept is always easier to predict, then even if $v$ represents the main-task features (e.g. the animal shape), the vector might be attributed to the spurious concept subspace, since it has a lower binary cross-entropy for the spurious concept label than the main task label.

To address this, we add the $\Delta$ term when testing for the second criterion mentioned in Section 3.3. By having a non-zero $\Delta$, we can account for the fact that one binary cross-entropy is always likely to be lower (or higher) than the other.

This naturally leads to the question how one should determine $\Delta$. We provide a simple heuristic. First, we optimize Equation 2 and obtain a first pair of spurious and main-task vectors, $\hat{v}_{\text{sp}}, \hat{v}_{\text{mt}}$. We compare the difference between these two orthogonal vectors through measuring the following:

$$d^*_{\text{sp,mt}} = \mathcal{L}_{\text{BCE}}(\hat{y}_{\text{sp}}^{(\hat{v}_{\text{sp}})}, y_{\text{sp}}) - \mathcal{L}_{\text{BCE}}(\hat{y}_{\text{mt}}^{\hat{v}_{\text{mt}}}, y_{\text{mt}}).$$

We measure the weighted average of this term, defined $\bar{d}^*_{w,\text{sp,mt}}$, for the validation set. This gives an indication if one set of labels is harder to predict than the other, and its value can be used to set $\Delta$.

In order to demonstrate the usefulness of $\Delta$ and the heuristic, consider the Toy dataset, outlined Section 4.1. In the original set-up, both sets of labels were equally hard to predict, since $\gamma_{\text{sp}} = \gamma_{\text{mt}} = 3$ for both. We change that for this section, and set $\gamma_{\text{sp}} = 6$, $\gamma_{\text{mt}} = 2$, making the spurious concept labels much more separable than the main-task labels.

In Figure 10 we show how JSE performs on a Toy dataset where the separability of the spurious and main-task labels differs. When we do not adjust $\Delta$ (e.g. keep it 0), we observe that both overall and worst-group accuracy drop once the correlation between the spurious and main-task features becomes high ($\rho = 0.9$). If we adjust $\Delta$ via our heuristic, this problem is avoided.

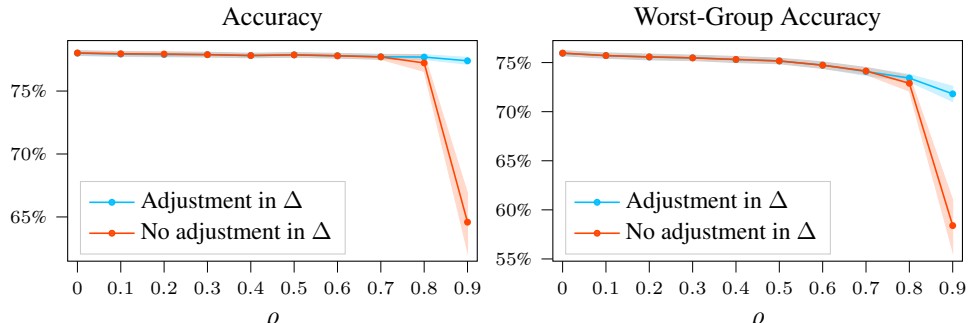

Figure 10: **Effect of adjusting $\Delta$ for JSE**: We plot the (worst-group) accuracy on a test set without spurious correlation, as a function of the spurious correlation in the training data. Each accuracy is obtained by averaging over 100 runs. The shaded area reflects the 95% confidence interval.

# E    USING JSE WHEN LIMITED SPURIOUS CONCEPT LABELS ARE AVAILABLE

In this section, we provide a comparison of JSE with other instance reweighing techniques such as JTT, GDRO, and GW-ERM, when there are limited spurious concept labels available. This is a relevant analysis, since it might be costly to acquire spurious concept labels in addition to the main-task labels.

Let $n_{sp}$ denote the number of spurious concept labels available, and $n$ the total number of datapoints. We presume that for all $n$ datapoints we have the main-task labels, but only know the spurious concept labels for a limited set $n_{sp} < n$.

In this experiment, we consider the Waterbirds and CelebA datasets. For JSE, we first estimate the spurious concept subspace $\mathcal{Z}_{sp}$, and then train an ERM model on the transformed embeddings with the $n$ main-task labels. GW-ERM and GDRO are fitted for the $n_{sp}$ datapoints where both the main-task and spurious concept labels are available. For JTT, we determine the hyper-parameters with the $n_{sp}$ datapoints for which the spurious concept label is available. For the other methods we keep the hyper-parameters the same as for experiments in Section 4.3.

Figure 11 and 12 show the reuslts of applying different instance reweighing methods to the problem of OOD generalization when there is a limited set of spurious concept labels available. Which method performs best appears to strongly depend on and the amount of available labels. For instance, with $n_{sp} = 1,000$, appears to be slightly better than the other methods for the Waterbirds dataset. However, when there are very few spurious concept labels, JSE performs similar to JTT for this dataset, as the estimation of the spurious concept subspace becomes harder with fewer labels. Noteworthy is the strong decrease in the performs of GW-ERM and GDRO, since these methods fit on much smaller datasets because they require knowledge of both the main-task and spurious concept label for each datapoint.

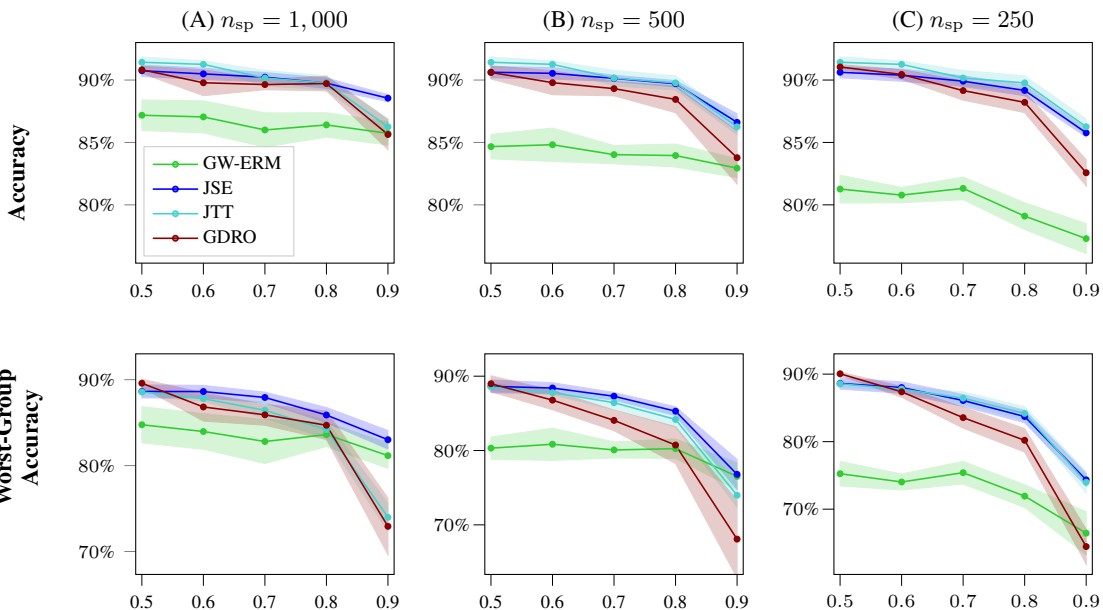

Figure 11: **OOD generalization for Waterbirds, different amounts of spurious concept labels:** We plot the (worst-group) accuracy on a test set without spurious correlation, as a function of the spurious correlation in the training set ($p_{\text{train}}(y_{\text{mt}} = y | y_{\text{sp}} = y)$), and the number of available spurious concept labels. Averages based on 20 runs. The shaded area reflects the 95% confidence interval.

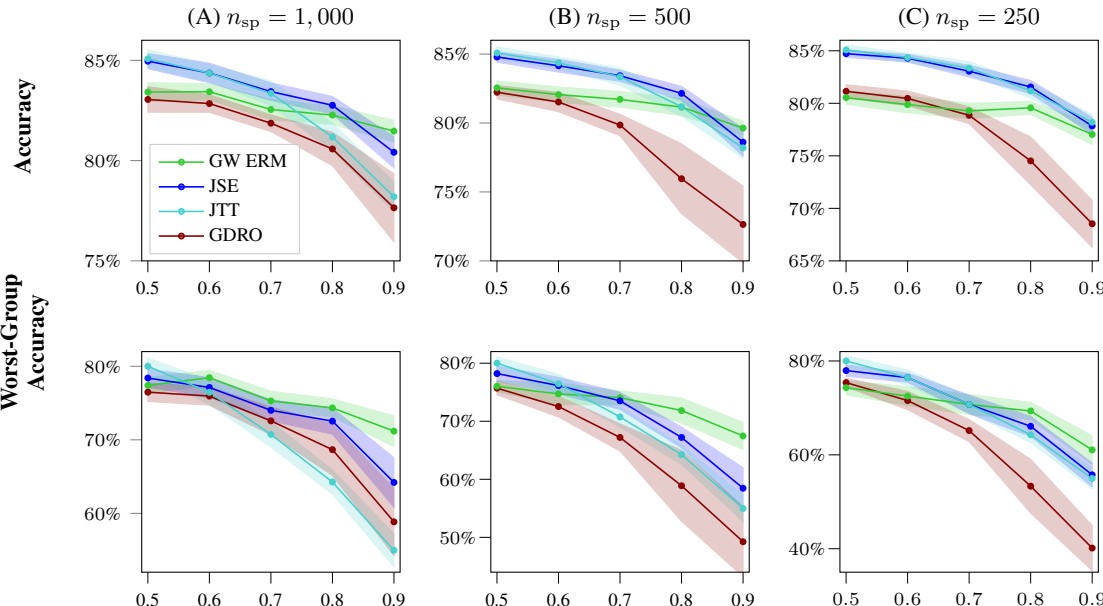

Figure 12: **OOD generalization for celebA, different amounts of spurious concept labels:** We plot the (worst-group) accuracy on a test set without spurious correlation, as a function of the spurious correlation in the training set ($p_{\text{train}}(y_{\text{mt}} = y | y_{\text{sp}} = y)$), and the number of available spurious concept labels. Averages based on 20 runs. The shaded area reflects the 95% confidence interval.

## F DETAILS ON DATASETS, MODELS, AND PARAMETER SELECTION

### F.1 DATASETS

**Toy:** For a given dataset size (e.g. $n =2,000$) the data is split into an 80% training and 20% validation set, and a test set of the same size is kept apart for evaluation. We describe the data-generating process of this dataset in Section 4.1.

**Waterbirds**: this dataset from Sagawa et al. (2020) is a combination of the Places dataset (Zhou et al., 2016) and the CUB dataset (Welinder et al., 2010). An 'water background' is set by selecting an image from the lake and ocean categories in the places dataset, and the 'land background' is set based on the broadleaf and bamboo forest categories. A waterbird/land is then pasted in front of the background. When creating new versions of the dataset, we change the $p(y_{\mathrm{mt}} = y|y_{\mathrm{sp}} = y)$, and keep the size of the training set at 4,775 samples, and 1,199 for the validation set. For the test set, we select 5,796 samples where $p(y_{\mathrm{mt}} = y|y_{\mathrm{sp}} = y) = 0.5$.

For this dataset when training ERM or adversarial removal, we sample in each batch such that $p(y_{\mathrm{mt}} = 1) = 0.5$. When training JSE, INLP or RLACE, we sample in each batch such that $p(y_{\mathrm{mt}} = 1) = 0.5$. When training ERM on the embeddings transformed by JSE, INLP or RLACE, we sample again such that in each batch $p(y_{\mathrm{mt}} = 1) = 0.5$.

$$y_{\mathrm{mt}} = 1, \quad y_{\mathrm{sp}} = 1 \qquad y_{\mathrm{mt}} = 1, \quad y_{\mathrm{sp}} = 0 \qquad y_{\mathrm{mt}} = 0, \quad y_{\mathrm{sp}} = 1 \qquad y_{\mathrm{mt}} = 0, \quad y_{\mathrm{sp}} = 0$$

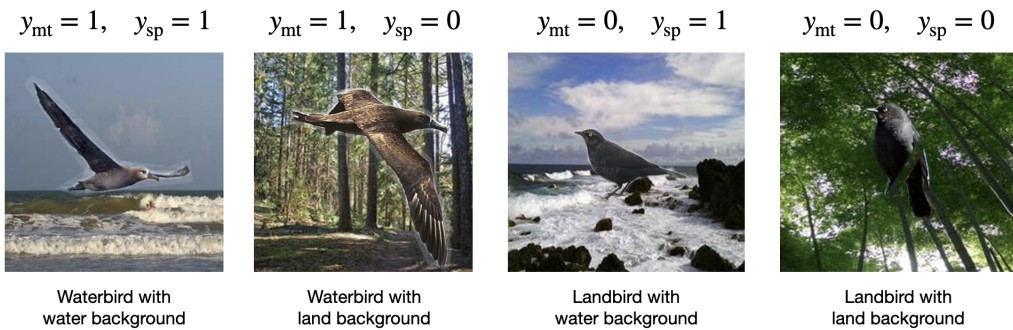

| Waterbird with water background | Waterbird with land background | Landbird with water background | Landbird with land background |

Figure 13: **Examples of the different images in the Waterbirds dataset**

**CelebA**: this dataset contains images of celebrity faces (Liu et al., 2015). The total size of the dataset is 202,599, from which we sample smaller versions in order to control the strength of the spurious correlation. For these smaller versions, we select 4,500 observations for the training set, 2,000 for the validation set, and 2,000 for the test set. We set the $p(y_{\mathrm{mt}} = y|y_{\mathrm{sp}} = y)$ for the training and validation set, while we set $p(y_{\mathrm{mt}} = y|y_{\mathrm{sp}} = y) = 0.5$ for the test set. In the smaller versions of the dataset, $p(y_{\mathrm{mt}} = 1) = 0.5$.

$$y_{\mathrm{mt}} = 1, \quad y_{\mathrm{sp}} = 1 \qquad y_{\mathrm{mt}} = 1, \quad y_{\mathrm{sp}} = 0 \qquad y_{\mathrm{mt}} = 0, \quad y_{\mathrm{sp}} = 1 \qquad y_{\mathrm{mt}} = 0, \quad y_{\mathrm{sp}} = 0$$

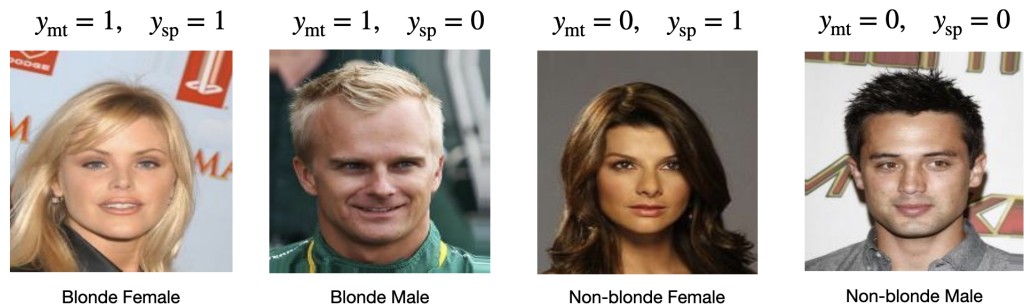

| Blonde Female | Blonde Male | Non-blonde Female | Non-blonde Male |

Figure 14: **Examples of the different images in the CelebA dataset**

**MultiNLI**: the MultiNLI dataset (Williams et al., 2018) contains pairs of sentences, with examples shown in Table 13. The sentences are pasted together with a [SEP] token in between. We change

the dependent variable to a binary label, with $y_{\mathrm{mt}} = 1$ indicating the first sentence (the premise) contradicting the second sentence (the hypothesis), and $y_{\mathrm{sp}}$ otherwise. The original dataset contains 206,175 pairs of sentences. We sample smaller versions of the dataset where $p(y_{\mathrm{mt}} = 1) = 0.5$ and change the $p(y_{\mathrm{mt}} = y|y_{\mathrm{sp}} = y)$ for the training and validation set. For the test set, we set $p(y_{\mathrm{mt}} = y|y_{\mathrm{sp}} = y) = 0.5$. The training set contains 50,000 datapoints and the validation and test set each contain 5,000 datapoints.

| $y_{\mathrm{sp}}$ | $y_{\mathrm{mt}}$ | Premise | Hypothesis |
|---|---|---|---|
| 0 | 0 | Conceptually cream skimming has two basic dimensions - product and geography. | Product and geography are what make cream skimming work !! |
| 1 | | One of our number will carry out your instructions minutely. | A member of my team will execute your orders with immense precision. |
| 0 | 1 | Fun for adults and children. | Fun for only children. |
| 1 | | This analysis pooled estimates from these two studies to develop a C-R function linking PM to chronic bronchitis. | The analysis proves that there is no link between PM and bronchitis !! |

Table 13: **Examples from sentence pairs in the MultiNLI dataset.**

For simplicity, we perform no data augmentation for any of the datasets. When training logistic regressions on the last-layer representations, we demean the data based on the mean of the training set. This is based on previous work from Chen et al. (2020), which states that demeaning is a necessary step before determining concept vectors.

## F.2 Models & Training procedure

**Models**: For the Waterbirds dataset, we use the Resnet50 architecture implemented in the `torchvision` package: `torchvision.models.resnet50(pretrained=True)`. More details on the model can be found in the original paper from He et al. (2016). PCA is applied to the embeddings of the layer of the architecture, to reduce the dimensionality from 2048 to 100.

For the MultiNLI dataset, we use the base BERT model implemented in the `transformers` package (Wolf et al., 2019): `BertModel.from_pretrained("bert-base-uncased")`. The model was pre-trained on BookCorpus, a dataset consisting of 11,038 unpublished books, as well as the English Wikipedia (excluding lists, tables and headers). More details on the model can be found in the original paper: Devlin et al. (2018). PCA is applied to the [CLS] embeddings, to reduce the dimensionality from 768 to 100.

**Training procedure**: For JSE, ERM INLP and RLACE, we use early stopping to prevent overfitting. If we observe no improvement on the validation set for a certain model after 5 epochs, the training procedure is stopped and the model with the lowest loss on the validation set is selected. We use stochastic gradient descent (SGD) with a momentum parameter of 0.9 for JSE, ERM INLP and RLACE, and train for a maximum of 50 epochs.

For finetuning the BERT model on MultiNLI we also use early stopping, and stop the procedure if we observe no improvement after 1 epoch. We use the Adam optimizer (Kingma & Ba, 2015) with the standard settings in Pytorch. When finetuning, we train for a maximum of 10 epochs, and use a batch size of 32, learning rate of $10^{-5}$, and a weight decay of $10^{-4}$.

For finetuning the Resnet50 model with adversarial removal, we again use early stopping, stopping if we observe no improvement after 1 epoch. We use the SGD optimizer, and train for a maximum of 20 epochs with a batch size of 128, learning rate of $10^{-3}$ and weight decay of $10^{-4}$.

## F.3 Implementation details & Parameter selection

We only assume access to a validation dataset that follows the same distribution as the training dataset. This means that the conditional probability $p(y_{\mathrm{mt}} = y|y_{\mathrm{sp}} = y)$ is the same across the training and validation set.

For experiments where we change the $p(y_{\mathrm{mt}} = y|y_{\mathrm{sp}} = y)$ in the training set, we do not select new parameters for each case, but select them based on the scenario where $p(y_{\mathrm{mt}} = y|y_{\mathrm{sp}} = y) = 0.9$.

For JSE, ERM, INLP, GDRO and JTT we set the batch size at 128. We also do so for RLACE, except in the case of the MultiNLI dataset - we only observed convergence for this dataset and method with a batch size of 512. For the experiments in Section E where we work with a lower number of spurious concept labels, we set the batch size to 64 to reflect the smaller dataset. After setting the batch size, we select the best combination of the learning rate and weight decay. For the learning rate, we assess the values $10^{-1}$, $10^{-2}$, $10^{-3}$ and $10^{-4}$. For the weight decay, we assess the values $0$, $10^{-3}$, $10^{-2}$, $10^{-1}$ and $1$. In the case of the Toy dataset we always set the weight decay to 0. For each method, we use the weighted binary cross-entropy on the validation set to measure performance. In the case of the toy, Waterbirds and CelebA dataset the performance is measured across 10 runs with a different seed. For the MultiNLI dataset, we measure it across 5 runs, each time finetuning the BERT model.

Below, we detail how the parameters were selected for each method, as well as implementation details for RLACE and adversarial removal. The selected combinations of the learning rate and weight decay can be found in Table 14.

**ERM**: We select the learning rate and weight-decay combination that has the best performance for the main-task labels. These parameters are also used when fitting a logistic regression on the transformed representations from JSE, INLP or RLACE. We keep the parameters the same for group-weighted ERM.

**INLP**: we select the combination of parameters that has the best performance for the spurious concept labels, based on the first spurious concept vector found by INLP. We continue projecting out the spurious concept vectors found by INLP until the accuracy of the spurious concept classifier is no better than a majority rule classifier. Whether or not the BCE is statistically significantly different from that of a majority rule classifier is tested via an $t-$test of the difference in the BCE's for both classifiers, where the critical value is determined based on $\alpha = 0.05$.

**JSE**: we select the combination of parameters that has the best performance for the spurious concept and main-task concept labels, weighing each equally. This performance is based on the first set of spurious and main-task concept vectors.

**RLACE**: we use the code from Ravfogel et al. (2022a), and run the algorithm for a maximum of 50,000 iterations. For the spurious concept classifier and optimizing the projection matrix, we use the same parameters as INLP. We optimize the projection matrix until the accuracy of the classifier is lower than 51%. Similar to Ravfogel et al. (2022a), in each case we find a matrix of rank 1.

**Adversarial removal**: The weight of the adversary loss is set to $\lambda = 1$. The entire architecture is finetuned, and the adversary is trained using the gradient reversal method (Ganin & Lempitsky, 2015). For the vision datasets, we observe that the accuracy of the adversary converges to below 55%, which is commonly accepted as success for the method. After the adversarial removal method, we apply standard ERM including PCA, as for the other methods.

For MultiNLI, we experimented extensively with hyper-parameters in order to get both a high accuracy on the main task, and the accuracy of the adversary to converge to below 55%. We did not observe this, even after lowering the weight of the adversary loss from 1 to $10^{-1}$ or even $10^{-2}$. We used the Adam optimizer when performing adversarial removal.

**GDRO:** we fix the hyper-parameter $\eta_g = 0.1$, similar to Idrissi et al. (2022). We grid-search the best combination of the learning rate, weight decay and the hyper-parameter of $C$, with possible values of 0, 1, 2, 3, 4, 5. We provided initial parameters for the GDRO model based on an fitted ERM model.

**JTT:** We grid-search the best combination of the learning rate, weight decay and hyper-parameter of $\lambda$, which is the weight for the misclassified samples.

| Dataset | Method | Learning Rate | Weight Decay |
|---------|--------|---------------|--------------|
| Toy | JSE | $10^{-2}$ | 0 |
| | ERM | $10^{-1}$ | 0 |
| | INLP | $10^{-1}$ | 0 |
| | RLACE | $10^{-1}$ | 0 |
| | GDRO | $10^{-3}$ | 0 |
| | JTT | $10^{-3}$ | 0 |
| Waterbirds | JSE | $10^{-3}$ | $10^{-3}$ |
| | ERM | $10^{-2}$ | $10^{-2}$ |
| | INLP | $10^{-1}$ | $10^{-3}$ |
| | RLACE | $10^{-1}$ | $10^{-3}$ |
| | ADV | $10^{-3}$ | $10^{-4}$ |
| | GDRO | $10^{-3}$ | $10^{-2}$ |
| | JTT | $10^{-3}$ | 1 |
| CelebA | JSE | $10^{-2}$ | $10^{-3}$ |
| | ERM | $10^{-2}$ | $10^{-2}$ |
| | INLP | $10^{-2}$ | $10^{-3}$ |
| | RLACE | $10^{-2}$ | $10^{-3}$ |
| | ADV | $10^{-3}$ | $10^{-4}$ |
| | GDRO | $10^{-3}$ | $10^{-3}$ |
| | JTT | $10^{-2}$ | $10^{-2}$ |
| MultiNLI | JSE | $10^{-2}$ | $10^{-2}$ |
| | ERM | $10^{-2}$ | 1 |
| | INLP | $10^{-2}$ | $10^{-3}$ |
| | RLACE | $10^{-2}$ | $10^{-2}$ |
| | GDRO | 1 | $10^{-4}$ |
| | JTT | $10^{-3}$ | $10^{-4}$ |

Table 14: **Selected combinations of learning rate and weight decay.** For the $C$ parameter of GDRO, we use the values $1, 2, 5$ and $5$ for respectively the Waterbirds, CelebA, multiNLI and Toy datasets. For the $\lambda$ parameter of JTT, we use the values of $5, 5, 10, 2$ for respectively the Waterbirds, CelebA, multiNLI and Toy datasets.

