# OpenReview forum: "Removing Spurious Concepts from Neural Network Representations via Joint Subspace Estimation"
_ICLR.cc/2024/Conference — Submitted to ICLR 2024_

### Official Review · Reviewer_YBBU · 2023-10-31

**Soundness:** 3 good
**Presentation:** 2 fair
**Contribution:** 2 fair
**Rating:** 5
**Confidence:** 4

**Summary:**

The paper focus out-of-distribution generalization in neural networks, and specifically spurious correlations. The methodology considered in the paper is to remove spurious concepts from the neural network's representation of data. To mitigate the weakness of the pervious methods along this line, which is the removal of important features, the authors propose an algorithm that identifies orthogonal subspaces in the neural network representation, separating spurious concepts from the main task. Experiments are done on a toy dataset as well as three benchmark data commonly used in research of spurious correlations.

**Strengths:**

(S1) The targeted problem of spurious correlation and OOD generalization is important.

(S2) The authors dedicated detailed discussions on their proposed method both in main paper and appendix.

**Weaknesses:**

(W1) The organization and clarity of this paper needs improvement to enhance its readability. For example, section 2 seems to be a discussion on the problem setting, but it's not clear what exactly is the data distribution model that the paper focus on upon reading the section. The assumptions are not properly and formally stated. A more structured discussion, with clear definitions and assumptions, would make it easier to understand the focus of this paper.

(W2) I would greatly appreciate more discussions on the experiment results.
1. It seems that ERM outperforms many of the baselines in both the toy data and real data. Given that the baselines are intended to enhance the model's performance on the worst-group accuracy, it would be beneficial if the authors could shed light on why ERM seems to have a superior performance than the presented baselines.
2. The selection of datasets—Waterbirds, CelebA, and MultiNLI—are frequently used to study spurious correlations. However, there exists a significant body of research [1-4] that specifically addresses these datasets and has demonstrated success in improving the worst-group accuracy. I understand that they are of different approach than this paper, but a proper discussion is needed on this line of methods and why they are not included in comparison.
3. There are some methods in the previous line of work that are mentioned in Appendix E, but I'm confused which method exactly is implemented as the baseline. By group weighted (GW) ERM, the authors cited papers on last layer retraining (DFR) as well as paper discussing group/class-balanced baselines that are not related to last layer retraining. These methods share similar concepts but have different methodologies and performances. I would appreciate a clearer explanation/citation of the exact compared baseline method.
4. In appendix E, it seems that the proposed method is not doing better than the baseline on the real datasets, but significantly outperforms it on the toy data. Is there a reasoning on why the toy data distribution specifically suits the proposed method, and would we be able to find similar distributions in real data?
5. The considered datasets typically have a very large value of $\rho$. For example, Waterbirds have $\rho=0.95$. The experiments of this paper stops at $\rho=0.9$, which does not seem like a conventional setting. It would be better to show improvement on the original dataset as well.

[1] "Just train twice: Improving group robustness without training group information." International Conference on Machine Learning. PMLR, 2021.

[2] "Environment inference for invariant learning." International Conference on Machine Learning. PMLR, 2021.

[3] "Correct-N-Contrast: a Contrastive Approach for Improving Robustness to Spurious Correlations." International Conference on Machine Learning. PMLR, 2022.

[4] "Robust Learning with Progressive Data Expansion Against Spurious Correlation." Advances in neural information processing systems. 2023.

**Edit**

Thank the authors for the rebuttal. While some of my concerns are addressed, I still hold concerns toward the experiment setting of this paper. Specifically, the ratio of spurious correlation is not standard across all datasets, not just a problem with Waterbirds. When considering spurious correlations, people (and many previous baselines) mostly focus on the setting where correlation is strong and therefore ERM fails significantly. The benchmark datasets therefore all hold a very large value of $\rho$, for example 0.97 with CelebA. These full datasets are not considered in this paper. Meanwhile, it is observable that JSE has a notable decrease in performance on the full dataset of Waterbird, leaving a gap between the baseline GroupDRO. Therefore, I maintain my score.

**Questions:**

See in weaknesses.

---

> ### Author Response · Authors · 2023-11-17
> **Detailed reply to reviewer YBBU**
>
> We are deeply grateful to reviewer YBBU for the careful consideration of the manuscript. The feedback has been invaluable for the revision of the manuscript.
>
> >The organization and clarity of this paper needs improvement to enhance its readability. For ... the focus of this paper.
>
> We have tried to improve the manuscript's readability, by more clearly structuring the different topics that we need to properly discuss and assess JSE. In Section 2 the focus is on the data distribution model (eq. 1) and the general approach and assumptions of (linear) concept removal. The assumptions that are specific to JSE can be found in Section 3. We have moved the discussion of OOD generalization from Section 2 to Section 4, as we realized this only becomes relevant in the context of the experiments we present in the manuscript.
>
> > I would greatly appreciate more discussions on the ... a superior performance than the presented baselines.
>
> Existing concept-removal methods tend to remove the spurious concept, but also other features that are correlated with the concept [1]. Because these main-task features are removed, a classifier that is trained on the transformed embeddings tends to do much worse than standard ERM. The purpose of Figure 3 (in the revised manuscript) is to show that JSE does not remove main-task features, as opposed to other concept-removal methods. To improve the clarity of this message, we moved the ERM results to the figure where JSE is compared to instance reweighing methods. In that figure, ERM can be seen as the default baseline.
>
> >The selection of datasets—Waterbirds, CelebA, and MultiNLI—are frequently ... discussion is needed on this line of methods and why they are not included in comparison.
>
> See the general comment (point 1) and the revised manuscript (in particular Section 4.3).
>
> >There are some methods in the previous line of work that are mentioned in Appendix E, but I'm confused which method exactly is implemented as the baseline. By group weighted (GW) ERM, ... of the exact compared baseline method.
>
> We thank the reviewer for raising this issue. We have added a description of the method in the main-text of the paper. In short, the last layer of the neural network is trained, and in each batch there is an equal probability to encounter a sample from each group – e.g. the probability of encountering a land bird on land is equal to that of a land bird on water. This is equivalent to the so-called 'RWG' baseline implemented in [2], but only applied to the last-layer - this why it was cited in the original manuscript.
>
> >In appendix E, it seems that the proposed method is not doing better than the baseline on the real datasets, but significantly outperforms it on the toy data. Is there a reasoning on why the toy data distribution specifically suits the proposed method, and would we be able to find similar distributions in real data?
>
> In the original manuscript we tried to clarify this in the appendix. In the revised manuscript, a reasoning is presented at the beginning of Section 4.3. JSE removes the spurious embedding, which makes it robust to changes at the more fundamental level of the conditional distribution $ p(y\_{\mathrm{mt}}| z\_{\mathrm{sp}}) .$ The other methods aim for robustness against changes in $ p(y\_{\mathrm{mt}}| y\_{\mathrm{sp}}).$ This explains why JSE outperforms the other methods for the Toy dataset, because in this setting the conditional distribution $p(y\_{\mathrm{mt}}| z\_{\mathrm{sp}})$ itself is changed for the OOD data.
>
> >The considered datasets typically have a very large value of $\rho$. For example, Waterbirds have $\rho=0.95$. The experiments of this paper stops at $\rho=0.9$, which does not seem like a conventional setting. It would be better to show improvement on the original dataset as well.
>
> We chose the range of $\rho$ to ensure consistency across all datasets, but we agree with the reviewer that in the specific case of Waterbirds $\rho=0.95$ should be included. We have done so in the revised manuscript in Section 4.3. We did not do this for the comparison with other concept-removal methods (Section 4.2), because at lower spurious correlations it was already clear that JSE outperforms them.
>
> [1] Abhinav Kumar, Chenhao Tan, and Amit Sharma. Probing classifiers are unreliable for concept removal and detection. In S. Koyejo, S. Mohamed, A. Agarwal, D. Belgrave, K. Cho, and A. Oh (eds.), Advances in Neural Information Processing Systems, volume 35, pp. 17994–18008. Curran Associates, Inc., 2022.
>
> [2] Badr Youbi Idrissi, Martin Arjovsky, Mohammad Pezeshki, and David Lopez-Paz. Simple data
> balancing achieves competitive worst-group-accuracy. In First Conference on Causal Learning
> and Reasoning, 2022.

---

### Official Review · Reviewer_YHWA · 2023-11-01

**Soundness:** 2 fair
**Presentation:** 2 fair
**Contribution:** 2 fair
**Rating:** 5
**Confidence:** 4

**Summary:**

This paper studies OOD generalization of neural network by removing spurious concepts with the proposed joint subspace estimation. The proposed method tries to separate spurious and main-task concepts in the embedding space, by jointly learning two low-dimensional orthogonal subspaces. It is an interesting work.

**Strengths:**

1.The motivation is clear. Separating and removing spurious from main-task concepts can prevent the model from using the spurious concept for main-task classification.
2.The organization and writing are well.
3.Extensive experiments are conducted to demonstrate its effectiveness.

**Weaknesses:**

1. unclear description about the orthogonality assumption. How to guarantee this assumption at any scene? Why linear subspaces are orthogonal? Why linear?
2.The novelty of the proposed jointly subspace estimation is limited and unclear.
3.Why logistic regression can separate spurious from main-task concept?

**Questions:**

see weaknesses

---

> ### Author Response · Authors · 2023-11-17
> **Detailed reply to reviewer YHWA**
>
> We thank reviewer YHWA for carefully reading our manuscript, and we are grateful for the useful questions and remarks.
>
> >unclear description about the orthogonality assumption. How to guarantee this assumption at any scene? Why linear subspaces are orthogonal? Why linear?
>
> *Why linear*: this is in line with the so-called linear subspace hypothesis, for which there is existing empirical evidence (see for instance [1,2]). Even if there is a non-linear representation of the concept, a linear classifier cannot use this information for the main-task if the concept is binary (see [3]).
>
> *Why orthogonal*: as stated in the paper, this is an assumption. We assume that the two concepts have a distinct representation in the embeddings – e.g. part of the embeddings are related to the background, and part are related to the bird type. Orthogonality is an intuitive way to formalize this distinctiveness (see for instance [4]). See also the general comment about the orthogonality assumption (point 4) and the revised manuscript.
>
>
> >The novelty of the proposed jointly subspace estimation is limited and unclear.
>
> We respectfully maintain a different perspective than the reviewer. Existing concept removal methods do not jointly determine the subspaces of main-task and spurious concepts. This makes them vulnerable to spurious correlations – as illustrated by our experiments. Furthermore, in our revised manuscript we have emphasized the advantages of concept-removal methods (interpretability, transfer learning, fairness/sensitive biases) as opposed to, for example, instance reweighing methods.
>
> >Why logistic regression can separate spurious from main-task concept?
>
> In principle, any linear classifier can be used. We chose logistic regression for convenience and because it allows for fast optimization, but if one believes that the best linear separation can be made by a support vector machine (SVM) or other generalized linear model, then they can use these to determinine the subspaces.
>
> An additional argument is that we apply logistic regression to the embeddings of the final layer. The decision boundaries of the original DNN are in fact linear in this space. Furthermore, the softmax output activation functions correspond to a logistic regression model from the embeddings to the outputs.
>
> [1] Tolga Bolukbasi, Kai-Wei Chang, James Y Zou, Venkatesh Saligrama, and Adam T Kalai. Man is to computer programmer as woman is to homemaker? debiasing word embed- dings. In D. Lee, M. Sugiyama, U. Luxburg, I. Guyon, and R. Garnett (eds.), Ad- vances in Neural Information Processing Systems, volume 29. Curran Associates, Inc., 2016.
>
> [2] Francisco Vargas and Ryan Cotterell. Exploring the linear subspace hypothesis in gender bias mit- igation. In Proceedings of the 2020 Conference on Empirical Methods in Natural Language Processing (EMNLP), pp. 2902–2913, Online, November 2020. Association for Computational Linguistics.
>
> [3] Shauli Ravfogel, Yoav Goldberg, and Ryan Cotterell. Log-linear guardedness and its implications. In Proceedings of the 61st Annual Meeting of the Association for Computational Linguistics (Vol- ume 1: Long Papers), pp. 9413–9431, Toronto, Canada, July 2023. Association for Computa- tional Linguistics
>
> [4] Park, K., Choe, Y. J., & Veitch, V. (2023). The Linear Representation Hypothesis and the Geometry of Large Language Models.

---

### Official Review · Reviewer_SKcn · 2023-11-01

**Soundness:** 3 good
**Presentation:** 3 good
**Contribution:** 2 fair
**Rating:** 5
**Confidence:** 4

**Summary:**

This paper addressed an out-of-distribution generalization problem under learning from a training dataset with spurious concepts by introducing Joint Subspace Estimation (JSE). Specifically, the authors assume that the main-task and spurious subspaces are orthogonal in the embedding space, and they proposed an algorithm for estimating these two subspaces simultaneously. In experiments, the authors showed that JSE outperforms the other concept removal methods (INLP, RLACE, and ADV) on the benchmark datasets. (modified Waterbirds, CelebA, and MultiNLI)

**Strengths:**

- The paper is well-written and easy to follow.
- The motivation for joint consideration of main and spurious subspace is sound and well reflected in the algorithm.
- The proposed JSE outperforms the concept removal baseline methods on the benchmark datasets.

**Weaknesses:**

- One of my concerns is regarding the experiment settings and baselines. The benchmark datasets used in this paper (Waterbirds, CelebA, and MultiNLI) are also used to evaluate debiased learning algorithms (GroupDRO, DFR, JTT, etc.). Since the settings are the same, JSE should be compared with these debiased learning methods. Alternatively, another option can be made by designing and showing experiment settings that differentiate concept removal methods from debiased learning algorithms.
- The suggested algorithm contains double for loops, which looks costly.
- The author leveraged PCA to reduce the computational cost, but it would bring out the information loss.
- Assuming the known $y_{sp}$ is not practical. Recently published debiased learning algorithms do not require spurious labels.

**Questions:**

- Is it possible to train and classify another waterbird dataset in which waterbirds are on the sand background, and landbirds are on the forest background (without $y_{sp}$ for both training and validation dataset) if we leverage the subspace information from the original waterbirds dataset?
- Does the orthogonality assumption always hold for every dataset and trained model?
- Could you compare the computational cost of JSE with INLP?
- Why $V_{sp}^\perp$ is used to train a last layer instead of $V_{mt}$? Is there an ablation study?

---

> ### Author Response · Authors · 2023-11-17
> **Detailed reply to reviewer SKcn**
>
> We thank reviewer SKcn for carefully reading the manuscript and the critical feedback, which has greatly helped us to improve the manuscript.
>
> >One of my concerns is regarding the experiment settings and baselines. ... from debiased learning algorithms.
>
> See the general comment (point 1) and the revised manuscript (in particular Section 4.3). On top of an empirical comparison, the revised manuscript also contains additional arguments in favor of concept removal methods (explainability, transfer learning, fairness/undesirable biases, robustness against feature shifts instead of label shifts).
>
> >The suggested algorithm contains double for loops, which looks costly.
>
> We understand that the reviewer raises this point. The double for-loop increases the computational cost for JSE, compared to methods such as INLP. However, we are only estimating a limited number of linear classifiers. JSE involves estimating $d\_{\mathrm{mt}} \times d\_{\mathrm{sp}}$ logistic regressions. In our experiments on realistic datasets, the dimensions of the respective subspaces were never found to be larger than 10. We address the issue of computational costs in the revised manuscript.
>
> >The author leveraged PCA to reduce the computational cost, but it would bring out the information loss.
>
> We understand the concern of the reviewer. At 100 components, a large percentage of the variation in the data ($>80\%$) is preserved (across all datasets). Furthermore, on realistic datasets the dimensions of the estimated subspaces were never found to be larger than 10, which is well below the 100 dimensions after PCA was conducted. It seems unlikely that the removed dimensions contain significant amounts of information about one of the concepts.
>
> The choice for using PCA was for computational convenience. We are currently running experiments of JSE without PCA and will add the results before the end of the rebuttal phase.
>
> >Assuming the known $y\_{\mathrm{sp}}$ is not practical. Recently published debiased learning algorithms do not require spurious labels.
>
> See the general comment (point 3) and the revised manuscript (appendix E).
>
> >Is it possible to train ... the original waterbirds dataset?
>
> We thank the reviewer for this useful suggestion. We believe this is certainly possible. Concept-removal methods can be used for transfer learning. In the current context, one could project out the background features (found with JSE) from the embedding space and do last-layer re-training on the new dataset.
>
> It may even be possible that re-training is not even needed. The Grad-CAM figure in the manuscript suggests that after JSE the model is actually looking at the correct input features (the bird, not the background). Presented with a new dataset, this should still be the case. We have included transfer learning as one of the potential benefits of concept-removal methods, but consider experiments using JSE for transfer learning as beyond the scope of the present manuscript.
>
> >Does the orthogonality assumption always hold for every dataset and trained model?
>
> We see that at high spurious correlation JSE has more difficulty with the text dataset (MultiNLI) than with the vision data, although it is still competitive with instance reweighing techniques and outperforming the other concept-removal methods. In the manuscript we elaborate on this and suggest that during the finetuning of BERT the main-task concept and spurious concept become overlapping in the [CLS] embedding, in line with observations from previous work [1]. This is potentially a sign that the orthogonality assumption does not always hold, or that the dimensions of the subspaces is so small that it is difficult to identify orthogonal subspaces within the subspaces. See also the general comment about the orthogonality assumption and the revised manuscript.
>
> >Could you compare the computational cost of JSE with INLP?
>
> Let’s presume that the spurious concept is represented by a $d\_{\mathrm{sp}}$-dimensional subspace, and the main-task concept by a $d\_{\mathrm{mt}}$-dimensional subspace. INLP involves estimating $d\_{\mathrm{sp}}$ logistic regressions, whereas JSE involves estimating $d\_{\mathrm{mt}}$ *  $d\_{\mathrm{sp}}$ logistic regressions. On realistic datasets the dimensions of the estimated subspaces were never found to be larger than 10.
>
> >Why $V\_{\mathrm{sp}}^\perp$ is used to train a last layer instead of $V\_{\mathrm{mt}}$? Is there an ablation study?
>
> In our original submission (as well as the updated version) there is an ablation study on this very question in Section C.2 of the appendix.
>
> [1] Fahim Dalvi, Abdul Rafae Khan, Firoj Alam, Nadir Durrani, Jia Xu, and Hassan Sajjad. Dis- covering latent concepts learned in BERT. In The Tenth International Conference on Learning Representations, ICLR 2022, Virtual Event, April 25-29, 2022. OpenReview.net, 2022

---

> ### Comment · Reviewer_SKcn · 2023-11-23
>
> Thank you for providing comprehensive responses and addressing the reviewer's concerns in the manuscript. While many of my major concerns have been resolved through the rebuttal and revisions, I have decided to maintain my initial score. My remained concerns are regarding the baselines. Notably, the performance of newly added baselines, particularly GDRO, appears lower than indicated in the existing literature. Additionally, the DFR results are still missing even though they are cited in the related works. I believe that addressing these issues in the revision could strengthen the manuscript. However, the current version may not meet the criteria for acceptance.

---

### Official Review · Reviewer_AJaV · 2023-11-02

**Soundness:** 2 fair
**Presentation:** 3 good
**Contribution:** 2 fair
**Rating:** 5
**Confidence:** 4

**Summary:**

The paper presents a method for removing spurious correlations in the latent representation by estimating the two orthogonal subspaces --- one associated with the spurious concept and the other with the main-task concept. The proposed method, Joint Subspace Estimation, use statistical test to identify directions in the embedding space associated with the shortcut and main task. The method is evaluated on the Waterbird, CelebA and MultiNLI dataset.

**Strengths:**

* The paper tries to address an important problem in ML --- detecting and mitigation spurious correlations. The paper is well-written and easy to follow.

* The idea of estimating orthogonal space is technically sound and novel.

* Results on the CelebA and Waterbird dataset shows that the method is able to disentangle spurious concept from the main concept. Visualization in ** Fig 6. ** confirms and validates this.

* The authors also evaluate an NLP dataset to demonstrate the method can work across different domains.

**Weaknesses:**

* The method depends on the availability of group labels (i.e., main task and spurious concept label), which is usually unavailable during training time.

* The method assumes the pixel corresponding to the main concept and the spurious concept doesn't overlap. This may not always hold true --- for e.g., if the main concept is the shape and the spurious concept is colour, the pixels can overlap.

* Definition of spurious concept in ** eqn 1** is not correct.
> label $y_mt$ and the spurious features $x_sp$ are independent

They are independent but correlated in the training data (spurious **correlations**)

> while the conditional and marginal distributions are same

I think this is incorrect. If both the conditional and marginal distributions are the same, the joint distribution will be the same too.

* Another major is the use of pre-trained Resnet50. Since it is trained on a large ImageNet data, it can extract features related to both main and spurious concepts.

* The method is benchmarked against enough baselines. Baselines should also include methods that do not use linear subspace projections/estimations, such as [1,2].




[1] Correct-N-Contrast: A Contrastive Approach for Improving Robustness to Spurious Correlations
[2] Just Train Twice: Improving Group Robustness without Training Group Information

**Questions:**

* How do you define main concept and spurious concept? For e.g., in CelebA, gender classification is a much harder problem than blond vs. non-blond hair.

---

> ### Author Response · Authors · 2023-11-17
> **Detailed reply to comments of reviewer AJaV**
>
> We are grateful to reviewer AJaV for examining the paper and for the detailed feedback. Below, we address the feedback point-by-point.
>
> >The method depends on the availability of group labels (i.e., main task and spurious concept label), which is usually unavailable during training time.
>
> See the general comment (point 3) and the revised manuscript (appendix E).
>
> >The method assumes the pixel corresponding to the main concept and the spurious concept doesn't overlap. This may not always hold true --- for e.g., if the main concept is the shape and the spurious concept is colour, the pixels can overlap.
>
> Thank you for raising this. JSE does not need the assumption of non-overlapping main-task and spurious input features. In fact, the sets of pixels corresponding to the main-task and spurious concept can even be identical. As long as the correlation between the pixel values and the respective labels is different, the DNN is able to learn distinct representations in the embedding space. The latter is in fact an assumption, which we explicitly state in Section 2 and which is based on existing literature [1,2,3]. We have corrected the text in the manuscript accordingly.
>
> >Definition of spurious concept in ** eqn 1** is not correct.
> >$$
> >\text{label  and the spurious features  are independent}
> >$$
> >They are independent but correlated in the training data (spurious correlations)
> >$$
> >\text{while the conditional and marginal distributions are same}
> >$$
> >I think this is incorrect. If both the conditional and marginal distributions are the same, the joint >distribution will be the same too.
>
> We respectfully hold a different view than the reviewer on this. Since $x\_{\mathrm{sp}}$ and $y\_{\mathrm{mt}}$ are correlated, they are necessarily dependent random variables. To clarify, $x\_{\mathrm{sp}}$ is not assumed to be independent of $y\_{\mathrm{mt}}.$ There is a causal relationship between $y\_{\mathrm{mt}}$ and $x\_{\mathrm{mt}},$ and because there is a dependency between $x\_{\mathrm{sp}}$ and $x\_{\mathrm{mt}},$ $x\_{\mathrm{sp}}$ is not independent of $y\_{\mathrm{mt}}.$
>
> Regarding the second statement, thank you for raising this issue. We understand the confusion, as we were referring to only the conditional distributions in eq. 1. Other conditional distributions indeed will change due to a change in the joint of $x\_{\mathrm{mt}}$ and $x\_{\mathrm{sp}}$. We now refer explicitly to the conditional distributions of eq. 1.
>
> >Another major is the use of pre-trained Resnet50. Since it is trained on a large ImageNet data, it can extract features related to both main and spurious concepts.
>
> Our method indeed assumes the presence of both spurious and main-task features in the embedding space. Previous work notes that many large neural networks learn both spurious and main-task features, even when trained on data with a spurious correlation [2]. If this holds, then our method should also work for a fine-tuned Resnet50 model (pre-trained on Imagenet data).
>
> >The method is benchmarked against enough baselines. Baselines should also include methods that do not use linear subspace projections/estimations, such as [1,2].
>
> See the general comment (point 1) and the revised manuscript (in particular Section 4.3).
>
> >How do you define main concept and spurious concept? For e.g., in CelebA, gender classification is a much harder problem than blond vs. non-blond hair.
>
> CelebA is a standard dataset for the problem of OOD generalization in the presence of spurious correlations. We followed the default definition of main-task and spurious concept, where gender is defined as the spurious concept.
>
> The hypothesis tests of JSE can actually account for a difference in predictability between the spurious or main-task concept. We show this in Appendix D.
>
> [1] Polina Kirichenko, Pavel Izmailov, and Andrew Gordon Wilson. Last layer re-training is suffi- cient for robustness to spurious correlations. In The Eleventh International Conference on Learn- ing Representations, ICLR 2023, Kigali, Rwanda, May 1-5, 2023. OpenReview.net, 2023.
>
> [2] Pavel Izmailov, Polina Kirichenko, Nate Gruver, and Andrew G Wilson. On fea- ture learning in the presence of spurious correlations. In S. Koyejo, S. Mo- hamed, A. Agarwal, D. Belgrave, K. Cho, and A. Oh (eds.), Advances in Neural In- formation Processing Systems, volume 35, pp. 38516–38532. Curran Associates, Inc., 2022.
>
> [3] Elan Rosenfeld, Pradeep Kumar Ravikumar, and Andrej Risteski. Domain-adjusted regression or: ERM may already learn features sufficient for out-of-distribution generalization. In NeurIPS 2022 Workshop on Distribution Shifts: Connecting Methods and Applications, 2022

---

> > ### Comment · Reviewer_AJaV · 2023-11-22
> > **Re**
> >
> > Thanks for addressing my concerns! I still have my concerns with using a pre-trained network to learn both main and spurious tasks. I am not sure if the hypothesis will hold true on a small model trained on a dataset with spurious correlation.

---

### Author Response · Authors · 2023-11-17
**General comment in reply to all 4 reviewers**

We are deeply grateful to all reviewers for their careful consideration of the paper. We also thank them for their acknowledgement of the relevance and novelty of our proposed JSE method, as well as their critical questions regarding our assumptions and the experiments we perform. Based on this criticism, much of which was shared by multiple reviewers, we have been able to substantially improve the manuscript. In this general comment we address the major points of criticism and how they are incorporated in the revised manuscript.

1. According to multiple reviewers, a comparison between JSE and state-of-the-art techniques that address OOD generalization in the presence of spurious correlations was missing. Although a comparison with group-weighted ERM (GW-ERM) was given in the appendix, other powerful instance reweighing methods like just train twice (JTT) and group distributional robust optimization (GDRO) should have been included.\
In the revised manuscript this comparison can be found in Section 4.3. We show that JSE is competitive with these methods, both for image and text data. As mentioned in the revised manuscript, it should be noted that contrary to the instance reweighing methods, the hyperparameters of JSE have been optimized for identifying the main-task and spurious concepts, and not for OOD generalization. If we had done so, the performance of JSE would have likely improved further.

2. As a result of the previous point of criticism expressed by the reviewers, we realized that the original manuscript did not pay justice to the advantages of concept-removal methods such as JSE. The focus was mainly on the application to OOD generalization, and not on the fact that in addition concept-removal methods make models better interpretable. This can be important in itself in terms of explainability, but it can also be used for transfer learning, for the removal of undesirable biases (e.g. gender, race). In this sense they have a considerable edge over techniques like GW-ERM, JTT and GDRO. We convey this message more clearly in the revised manuscript, particularly in Section 1.\
In Section 4 we also put more emphasis on the fact that concept-removal methods make models robust to shifts in $p(z\_{\mathrm{sp}}, z\_{\mathrm{mt}})$ instead of $p(y\_{\mathrm{sp}},  y\_{\mathrm{mt}}).$ This is an additional argument in favor of JSE, as previous work notes that simply balancing based on the group labels can be insufficient to remove information about certain concepts [1].

3. Another shared point of criticism was the reliance of JSE on the availability of spurious concept labels. In Appendix E of the revised manuscript we run the OOD generalization comparison with GW-ERM, JTT and GDRO for smaller numbers (1000, 500, 250) of available spurious concept labels. We find that JSE can perform well in this regime. Compared to GW-ERM and GDRO, the decrease in generalization performance for fewer spurious concept labels is less. For very few available labels (e.g. 250) JTT becomes competitive with JSE. This can be explained by the fact that JTT only needs the labels for a small validation set that is used to tune the hyperparameters.

4. A final shared point of criticism is JSE's orthogonality assumption of the main-task and spurious subspaces. This is an operational assumption needed in order to jointly estimate both subspaces, and partly justified by the performance of JSE on multiple realistic datasets.\
In Section 3.1 of the revised manuscript we have included an additional argument. Even if the subspaces themselves are not orthogonal, their high-dimensional nature provides enough degrees of freedom to identify orthonormal sub-subspaces that cover significant parts of the respective subspaces (in terms of predictiveness of the labels). In other words, even if the spaces are not orthogonal, large fractions of the main-task and spurious information can be captured. We further illustrate this in Section B of the appendix for the Toy dataset.

We hope that these considerations and the revised manuscript address the main concerns of the reviewers in a satisfactory manner. We thank them again for their thorough feedback. We post comments to each individual reviewer with a detailed reply to their specific questions and remarks.

[1] Wang, T., Zhao, J., Yatskar, M., Chang, K.-W., & Ordonez, V. (2019, October). Balanced Datasets Are Not Enough: Estimating and Mitigating Gender Bias in Deep Image Representations. International Conference on Computer Vision (ICCV).

---

### Meta-Review · Area_Chair_QBNh · 2023-12-06

**Metareview:**

The paper proposes a novel method for removing spurious concepts from neural network representations. The idea is to jointly estimate linear subspaces for the spurious and task-relevant concepts. The two subspaces are assumed to be orthogonal, and both target and spurious labels are needed for estimating the subspaces. The authors show promising results on spurious correlation benchmarks compared to baselines in spurious correlations and concept removal literature.

## Strengths

The proposed method is interesting, simple and practical. The empirical results are promising.

## Weaknesses

The main concern that multiple reviewers expressed is that the evaluation is non-standard, and that baselines are not clearly comparable to prior work. This is an important issue. The paper uses standard benchmarks, but modifies them by considering varying spurious correlation strengths. This is an interesting ablation, but it makes it impossible to directly compare the results to those reported by prior work. In particular, reviewers suggested that the baseline might be underperforming.

The authors use an unusual setting where they do not train the base model on the target task and just use a pretrained model. While this is a valid setting, it again prevents comparison to prior work.

## Notes

I wanted to make a couple small notes based on the discussion:
- The authors mentioned that according to [[Idrissi et al]](https://proceedings.mlr.press/v177/idrissi22a.html) group-reweighted ERM is competitive with GroupDRO. I believe this result only holds under very heavy tuning of hyper-parameters that [Idrissi et al] did. For example, compare RWG and Group DRO in Fig. 1 of [[Izmailov et al]](https://arxiv.org/abs/2210.11369)
- I believe, for [[DFR]](https://arxiv.org/abs/2204.02937) it was reported that it is important to subset the data to equal groups rather than subsample the data per-batch. Similar observation is reported in [[Idrissi et al]](https://proceedings.mlr.press/v177/idrissi22a.html), SUBG vs RWG, Fig. 1.

**Justification For Why Not Higher Score:**

I believe it is important for a paper proposing a method for reducing spurious correlations to report results that can be directly compared to prior work. Multiple reviewers flagged the issue with the evaluation. Generally, the reviewers voted unanimously in favor of rejecting the paper.

**Justification For Why Not Lower Score:**

N/A

---

### Decision · Program_Chairs · 2024-01-16

Reject